# scooby: modeling multimodal genomic profiles from DNA sequence at single-cell resolution

Johannes C. Hingerl [1,2,9], Laura D. Martens [1,3,9], Alexander Karollus[1,2], Trevor Manz [4], Jason D. Buenrostro [5,6], Fabian J. Theis [1,3,7] & Julien Gagneur [1,2,3,8] ✉

Understanding how regulatory sequences shape gene expression across individual cells is a fundamental challenge in genomics. Joint RNA sequencing and epigenomic profiling provides opportunities to build models capturing sequence determinants across steps of gene expression. However, current models, developed primarily for bulk omics data, fail to capture the cellular heterogeneity and dynamic processes revealed by single-cell multimodal technologies. Here, we introduce scooby, a framework to model genomic profiles of single-cell RNA-sequencing coverage and single-cell assay for transposase-accessible chromatin using sequencing insertions from sequence at single-cell resolution. For this, we leverage the pretrained multiomics profile predictor Borzoi and equip it with a cell-specific decoder. Scooby recapitulates cell-specific expression levels of held-out genes and identifies regulators and their putative target genes. Moreover, scooby allows resolving single-cell effects of bulk expression quantitative trait loci and delineating their impact on chromatin accessibility and gene expression. We anticipate scooby to aid unraveling the complexities of gene regulation at the resolution of individual cells.

Modeling the relationship between genetic sequence and measured molecular traits is an effective strategy for uncovering the genetic basis of gene regulation and complex traits[1]. Deep learning modeling can be used to identify genetic determinants of genomic readouts and to predict the effects of genetic variants directly from DNA sequence[2–8]. With the advent of single-cell technologies, sequence-based models of single-cell omics assay were developed aiming at unraveling sequence determinants of cell-state-specific regulation and cell-fate decisions[9–13]. Early work modeled local chromatin accessibility by predicting data from the single-cell assay for transposase-accessible chromatin using sequencing (scATAC-seq) with convolutional neural networks operating on small input sequences (1 kb)[9,11]. While scATAC-seq provides valuable insights into chromatin state and transcription factor (TF) binding, gene expression remains the ultimate functional readout of regulatory activity. Therefore, local predictors of binarized or pseudobulked gene expression have been proposed[10,13], yet their sequence context is limited and fails to capture cell-type-specific gene regulation[12].

[1]School of Computation, Information and Technology, Technical University of Munich, Munich, Germany. [2]Munich Center for Machine Learning, Munich, Germany. [3]Computational Health Center, Helmholtz Center Munich, Neuherberg, Germany. [4]Department of Biomedical Informatics, Harvard Medical School, Boston, MA, USA. [5]Gene Regulation Observatory, Broad Institute of MIT and Harvard, Cambridge, MA, USA. [6]Department of Stem Cell and Regenerative Biology, Harvard University, Cambridge, MA, USA. [7]School of Life Sciences Weihenstephan, Technical University of Munich, Munich, Germany. [8]Institute of Human Genetics, School of Medicine and Health, Technical University of Munich, Munich, Germany. [9]These authors contributed equally: Johannes C. Hingerl, Laura D. Martens. ✉e-mail: gagneur@in.tum.de

To address this, the recently presented seq2cells[12] model adapts Enformer[7], a state-of-the-art sequence-based model for gene expression prediction trained on thousands of bulk omics assays to infer single-cell gene expression counts from 200 kb of sequence. However, seq2cells only models gene expression, relies on potentially ambiguous transcription start site (TSS) annotations[8], and models each cell with a separate output track, leading to computational intractability for large datasets.

To overcome these limitations, we present scooby, which jointly models scATAC-seq and single-cell RNA-sequencing (scRNA-seq) genomic profiles from sequence without the need for annotations and easily scales to large datasets. scooby builds upon Borzoi[8], a recently released bulk omics model that uses RNA-seq coverage as an annotation-free representation of gene regulation. Using Borzoi as a foundation model, we equip it with a cell-specific decoder, and fine-tune its sequence embeddings to adapt it to the single-cell setting. We call this adapted model 'scooby', a nod to both its single-cell focus and its canine-named model ancestry. We demonstrate scooby's capabilities to accurately model genomic profiles at single-cell resolution, identify lineage-specific regulators and their putative target genes, and delineate cell-type-specific variant effects on a multiome hematopoiesis dataset, benchmarking it against state-of-the-art models.

## Results

### scooby enables modeling of genomic profiles at single-cell resolution

Here, we present scooby, which models single-cell accessibility and expression profiles from DNA sequence (Fig. 1a). Our model builds upon Borzoi, a state-of-the-art sequence-based model for RNA-seq coverage prediction, and leverages its trained convolutional and transformer-based architecture to extract informative sequence embeddings at 32-bp resolution. To tailor these embeddings to individual single-cell datasets and achieve single-cell resolution profile predictions, scooby introduces two key innovations (Fig. 1a and Extended Data Fig. 1).

First, to enable scooby to efficiently adapt to dataset-specific features, we fine-tuned its sequence embeddings using low-rank adaptation (LoRA)[14], a parameter-efficient fine-tuning strategy. Following the LoRA approach, we kept pretrained weights frozen and added trainable low-rank matrices into the transformer and convolutional layers (Methods). Advantageously, these matrices can be merged into the existing weights after training, resulting in no overhead during model inference[14]. We reasoned that this would allow scooby to capture effects of regulatory sequences relevant to cell states that are absent from or weakened in the bulk data Borzoi was trained on, but also to adjust to characteristics inherent to single-cell assays, such as the 3′ coverage bias common in scRNA-seq[15].

Second, we implemented a lightweight decoder for gene expression and accessibility prediction at single-cell resolution. To this end, scooby leverages low-dimensional, multiomic representations of cell states, in this case derived from Poisson-MultiVI[16,17], to decode the fine-tuned sequence embedding generated by Borzoi in a cell-specific manner. This design differs from approaches that rely on separate output heads for each cell[12], whose number of parameters scales with the number of cells and cannot, by design, effectively leverage similarities between cells. For efficient analysis of large single-cell datasets with scooby, we developed an accessible workflow by adapting SnapATAC2.0 (ref. 18) to store single-cell profiles in the widely used AnnData[19,20] format, which facilitates memory-efficient model training (Extended Data Fig. 2, Supplementary Fig. 1 and Methods).

We ensured robust evaluation by following the same sequence-level train and test splits as our underlying foundation model Borzoi. Moreover, genes and scATAC-seq peaks overlapping with validation or test regions were excluded from the input data used to generate the single-cell embeddings to avoid data leakage.

We trained scooby on a 10x Single Cell Multiome dataset (joint single-nuclei RNA-seq and ATAC-seq) comprising 63,683 human bone marrow mononuclear cells[21] (NeurIPS dataset; Fig. 1b and Methods) across eight NVIDIA A40 GPUs for 2 days until convergence. A typical example of the model prediction for a single cell is shown in Fig. 1c for the *SLC25A37* locus. Despite the inherent sparsity of single-cell data, the predicted profile for this single cell captures its expected profile as indicated by the empirical averages over its 100 neighboring cells. Moreover, the model precisely predicted the localization of scRNA-seq signal at the 3′ end of transcripts. This shows that scooby successfully adapted to the scRNA-seq assay, despite Borzoi being originally trained to model full-length RNA-seq coverage. Importantly, the model effectively captured the differential regulation of the *SLC25A37* locus, accurately inferring lower expression in a megakaryocyte–erythroid progenitor cell compared to an erythroblast cell and highlighting distinct accessibility patterns between these cells (Fig. 1c).

To assess the performance of scooby globally, we first computed the Pearson correlation between observed single-cell profiles and the predicted profiles, on a logarithmic scale for a random subset of cells of each cell type across all test sequences (Fig. 1d and Methods). Assessing optimal performance for scRNA-seq and scATAC-seq profile prediction is challenging due to the lack of a true ground truth. As a practical upper bound, we used the correlation between individual cell profiles and the average profile of their 100 nearest neighbors, representing a smoothed, less noisy signal. Additionally, we compared each single-cell profile to its corresponding pseudobulk profile. We found scooby's predictions to improve correlations compared to the corresponding pseudobulk profiles for both scRNA-seq (mean Pearson correlation = 0.15 versus 0.09) and scATAC-seq (mean Pearson correlation = 0.11 versus 0.08; Fig. 1d), yet these remained below the upper bound. However, when comparing scooby's single-cell predictions to the 100-nearest-neighbor average, correlations significantly increased (0.63 and 0.70 for scRNA-seq and scATAC-seq, respectively; Fig. 1d), indicating that scooby effectively captures the underlying signal in single-cell profiles despite their sparsity. Collectively, these results suggest that, while further model advances are possible, scooby models cell-specific regulation with increased finesse compared to pseudobulk averaging.

### scooby precisely captures cell-type-specific gene expression

Given that marker genes exhibit distinct expression patterns across cell types, we reasoned that accurate prediction of these genes would provide initial evidence for scooby's capacity to capture cell-state-specific gene expression. To derive gene expression counts, we summed the predicted scRNA-seq coverage across exons for each cell and compared these predicted counts to the observed single-cell counts (Fig. 2a and Methods). The model accurately captured cell-state-specific expression levels for marker genes unseen during training, even for small cell populations (Fig. 2b). In particular, it accurately recalled expression profiles of the markers *ANK1*, *DIAPH3*, *SLC25A37* and *AUTS2* that distinguish different cell types of erythroid differentiation (MK/E progenitors, (pro-)erythroblasts, normoblasts).

For a quantitative analysis, we next evaluated scooby's performance at predicting pseudobulked gene expression profiles for each cell type (Fig. 2c). We calculated the log-fold change between predictions and the observed pseudobulk expression levels for each gene within each annotated cell type. Across all cell types, we observed a mean Pearson correlation ranging from 0.82 to 0.88 (mean Pearson *R* = 0.86; Fig. 2d and Extended Data Fig. 3), demonstrating accuracy comparable to the original Borzoi model trained on bulk RNA-seq data (0.86 mean Pearson *R*). To assess the extent to which scooby captures differential expression patterns, we calculated the correlation after subtracting both the gene-wise and cell-type-wise means on a logarithmic scale (Fig. 2c). This analysis, which focused on deviations from the global mean expression, yielded a Pearson correlation

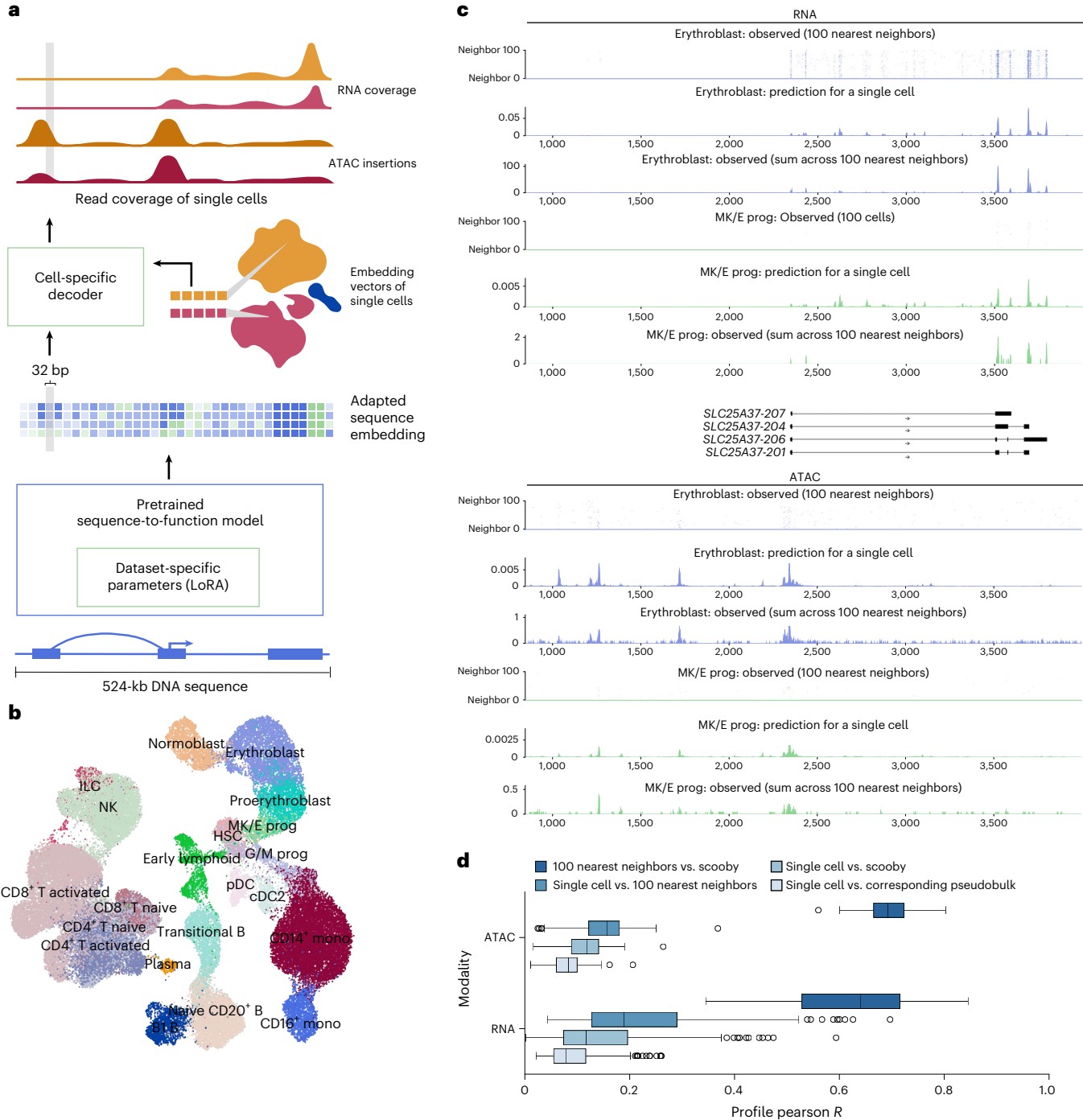

**Fig. 1 | scooby accurately predicts cell-state-specific expression and accessibility profiles from single-cell data. a**, scooby integrates a pretrained sequence-to-profile model with a cell-state-specific decoder to model genomic profiles at single-cell resolution. The pretrained model is fine-tuned on the target dataset using a parameter-efficient strategy, generating an adapted sequence embedding at 32-bp resolution. The cell-state-specific decoder takes this sequence embedding together with embedding vectors of single cells as input to predict scATAC-seq insertion and scRNA-seq coverage profiles at single-cell level. **b**, Uniform manifold approximation and projection (UMAP) visualization of the 10x multiome NeurIPS bone marrow dataset[21] integrated with Poisson-MultiVI, colored by cell type. **c**, Representative example of predicted and observed gene expression (top) and accessibility (bottom) profiles of an erythroblast and a megakaryocyte–erythroid progenitor cell and its 100 nearest neighbors at the *SLC25A37* locus (part of the test set). **d**, Distribution of the correlation between

predicted and observed profiles on test sequences (*n* = 210 representative cells; Methods). Box plots showing the distribution of Pearson *R* values for different comparisons of single-cell ATAC-seq and RNA-seq profiles. Comparisons include single cells versus corresponding pseudobulk, single cells versus scooby, single cells versus 100 nearest neighbors and 100 nearest neighbors versus scooby. All pairwise comparisons per assay are statistically significant (two-sided Wilcoxon rank-sum test, ATAC: *P* = 3 × 10$^{-36}$, 1 × 10$^{-35}$, 7 × 10$^{-34}$. RNA: *P* = 3 × 10$^{-36}$, 4 × 10$^{-36}$, 5 × 10$^{-20}$). In all box plots, the central line denotes the median, boxes represent the interquartile range and whiskers show the distribution except for outliers. Outliers are all points outside 1.5 times the interquartile range. B, B cell; T, T cell; Mono, monocyte; prog, progenitor; HSC, hematopoietic stem cell; ILC, innate lymphoid cell; Lymph, lymphoid; MK/E, megakaryocyte and erythrocyte; G/M, granulocyte and myeloid; NK, natural killer cell; cDC2, classical dendritic cell type 2; pDCs, plasmacytoid dendritic cells.

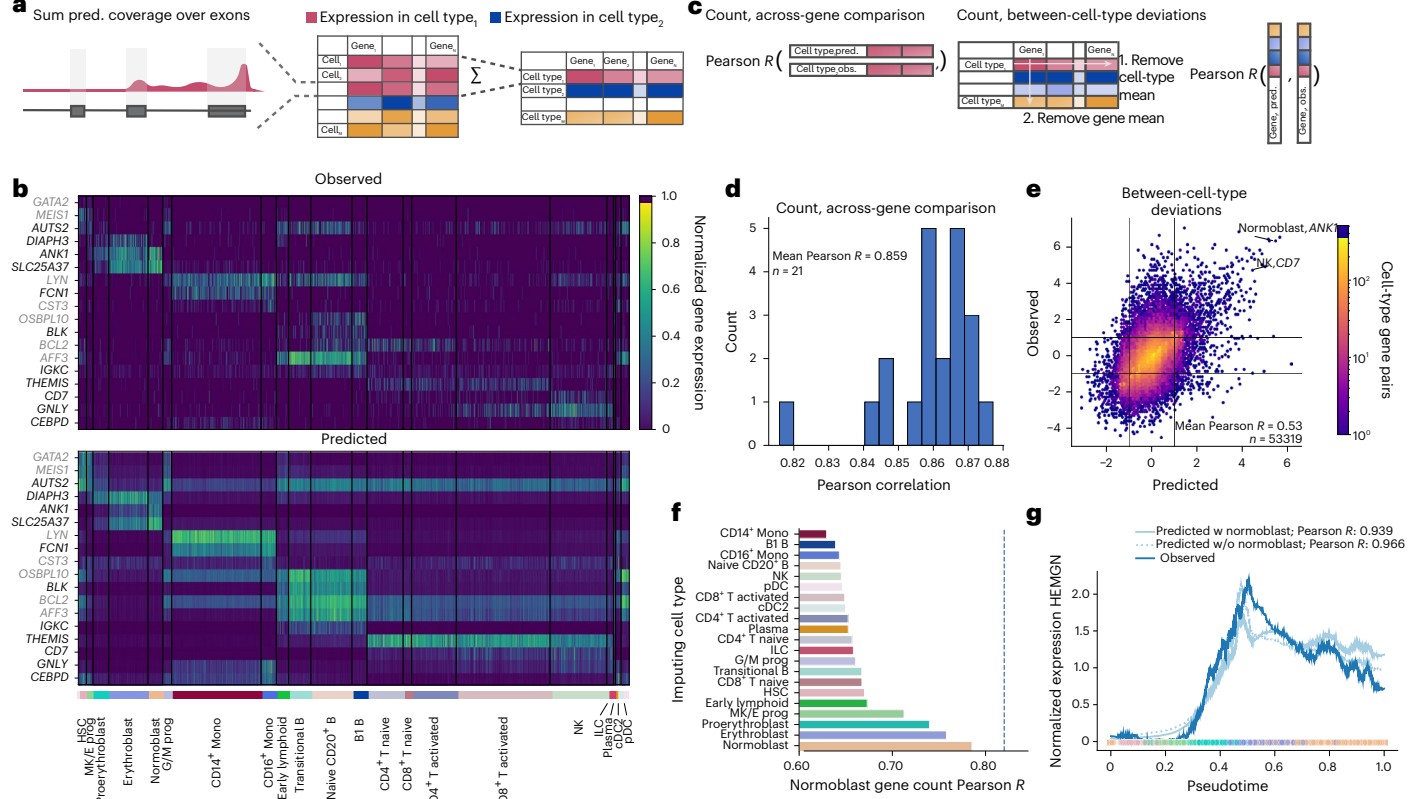

**Fig. 2 | scooby accurately models cell-type-specific gene expression counts and generalizes to unseen cell states. a**, Predicted and observed profiles are aggregated into a gene expression count matrix by summing coverage over exons. We obtain pseudobulk counts by summing over all predictions of every cell for each cell type. **b**, Normalized gene expression matrix (Methods) for cell-type-specific genes, observed (top) and predicted (bottom). Each row is a marker gene from test (black) or validation (gray), and each column is a randomly selected cell. Cells are grouped by cell type (bottom track) **c**, We evaluate scooby's performance using two metrics: the correlation between predicted and observed gene expression counts within each cell type (left) and the model's ability to capture cell-type-specific deviations of gene expression to gene mean (right). **d**, Distribution of gene-level Pearson correlation between log-transformed predicted and observed counts of scRNA-seq reads

overlapping exons across cell types. **e**, Predicted against measured between-cell-type deviations of gene expression. Exemplarily highlighted combinations of marker gene and cell type show strong deviations from the mean expression level. **f**, Across-gene Pearson correlation between log-transformed predicted and observed normoblast gene expression counts using an ablated model that was not trained on normoblast cells. Each bar corresponds to predictions done using the single-cell embeddings of cells of a different cell type. **g**, Mean-normalized observed and predicted gene expression of *HEMGN* along the diffusion pseudotime axis representing erythropoietic differentiation[22]. Both the full and the no-normoblast model accurately recapitulate the expression dynamics. Dots are colored by cell type, and lines are smoothed with a rolling mean (window size, 200 cells).

of 0.54, indicating that scooby successfully recapitulated a substantial portion of the biological variation in gene expression across cell types (Fig. 2e). Notably, scooby substantially outperformed the count-based seq2cells model retrained on the NeurIPS dataset on shared Enformer and Borzoi test genes, with mean correlation across genes increasing from 0.77 to 0.87 and mean correlation across cell types increasing from 0.43 to 0.55 (Methods and Extended Data Fig. 4).

We next performed some comparative analyses to understand the basis of scooby's performance (Extended Data Fig. 5). A scooby model trained on the scRNA-seq data and using embeddings purely derived from this modality performed worse than the multiomic model (across-gene Pearson $R = 0.848$, across cell types Pearson $R = 0.496$). Nevertheless, these results remain better than those of the seq2cells model and indicate that the increased performance of scooby is not limited to the fact it can leverage more data modalities.

To assess the contribution of dataset-specific fine-tuning, we tested a scooby variant without LoRA fine-tuning, but instead directly feeding Borzoi embeddings into the cell-specific decoder. This resulted in decreased prediction accuracy, particularly for relative expression between cell types (across cell types Pearson $R = 0.501$), highlighting the importance of fine-tuning all layers of the model to the dataset

(Extended Data Fig. 5). Consistent with that observation, we found that simpler models which only had cell-state-specific decoders on top of Borzoi's human RNA-seq predictions and on top of all Borzoi's predicted human tracks also performed notably worse (Extended Data Fig. 5). Furthermore, we designed a comparable model on pseudobulked profiles, which are predicted in a multi-task fashion while keeping the rest of the architecture the same. Still, scooby performed on par or slightly better on both cell-type-level metrics, showing that the scooby architecture does not trade cellular resolution for performance.

### scooby predicts gene expression dynamics of unseen cells

A salient feature of scooby over seq2cells is that cell-specific predictions are achieved using a single-cell embedding representation instead of modeling every single cell as a distinct task. This design enables application to unseen cells within similar cell states, which could be used in scenarios like atlas mapping, where new datasets are projected onto a reference. However, we do not expect scooby to generalize to drastically different cell types beyond its training domain.

To assess robustness of the single-cell decoder to slightly different cell states, we evaluated a model where all normoblast cells, constituting the terminal cell type of the erythroid lineage, were withheld

both during construction of the single-cell embedding and training of scooby (Methods). Remarkably, using normoblast embeddings that were projected into the learned embedding after training yielded predictions with an accuracy close to the model trained on the full dataset (0.79 Pearson $R$ compared to 0.81 for the model trained with normoblasts). Moreover, the best predictions were obtained using the normoblast embeddings as input to the decoder, followed by using embeddings of closely related cell types, indicative of scooby learning meaningful representations in embedding space that can be used to interpolate between different cell states (Fig. 2f).

Building on this observation, we investigated whether scooby's capacity to generalize to unseen but related cell states extends to capturing the continuous gene expression changes that occur during differentiation. As a case study, we considered hemogen (*HEMGN*), a gene known to be upregulated during erythroid differentiation[22] that was part of the sequences held out during model training. Using diffusion pseudotime[23] to order cells along the erythroid trajectory, we compared *HEMGN* expression dynamics predicted by both the full scooby model and the model trained without normoblasts (Fig. 2g). Both models, including the one trained without normoblasts, accurately recapitulated the regulation of *HEMGN* along the erythroid lineage (0.939 Pearson $R$ for the full model and 0.966 for the ablated model).

Altogether, these results indicate that scooby can be applied to investigate unseen, but related cell states and continuous regulatory programs similar to the ones observed during model training. This makes it suitable to use as a tool in reference atlas integration workflows where one might want to interpret novel datasets with related cell states by mapping it to a known reference.

## TF motif effect scoring allows investigating TF activity

Having established scooby's capability to predict cell-state-specific gene expression, we next sought to understand the sequence determinants influencing its predictions. Given the central role of TFs in regulating gene expression, we aimed to identify TFs that drive lineage-specific gene expression predictions. To quantify the importance of TFs on gene expression, we introduced a TF motif effect score, which measures the impact of mutating TF binding sites on predicted gene expression in single cells. Specifically, we focused our analysis on 83 TFs that are significantly differentially expressed between cell types of the bone marrow dataset. For each TF, we used established position weight matrices (PWMs)[24] to map candidate TF binding sites located within 524 kb centered on the gene body of the 3,681 genes differentially expressed across the dataset (Fig. 3a and Methods). We mutated all matching sites in silico at the same time by substituting them with random sequences. Cell-level TF motif effect scores were defined as the log-fold change of scooby's predicted expression between reference sequence and in silico mutated sequences, averaged across genes. TF motif effect scores are directional, being positive for activators and negative for repressors.

To assess the reliability of scooby's TF motif effect scores, we compared their agreement with observed cognate TF expression, as a proxy for true TF activity (Methods). We benchmarked against chromVAR[25] and scBasset[9], two widely used sequence-based methods for inferring TF activity from scATAC-seq data by comparing the correlation of their scores with TF expression. We found that scooby's TF motif effect scores correlate significantly better with gene expression than those of chromVAR ($P = 5.4 \times 10^{-9}$, Wilcoxon two-sided; Fig. 3b) and scBasset ($P = 0.04$, Wilcoxon two-sided; Extended Data Fig. 6a). Remarkably, training scooby only with scRNA-seq data led to TF motif effect scores on par or better than the two alternative methods chromVAR and scBasset, which use scATAC-seq data (Fig. 3c, Extended Data Fig. 6b and Methods). This result indicates that scooby could alleviate the need for scATAC-seq data for the purpose of TF activity inference.

Having established TF motif effect scores, we next leveraged them to investigate the regulatory role of activating or repressing TF

sequence elements on gene expression in a cell-type-specific manner. We observed scooby to recapitulate the importance of known motifs for cell types of the main hematopoietic lineages (Fig. 3d and Extended Data Fig. 7a). For example, the GATA1 motif family exhibited the highest score in erythroblasts[26], the EBF1 motif in B1 B cells[27], the C/EBP motif family in monocytes[28] and the RUNX motif family in T cells[29]. The SOX motif family, containing a TF known to drive multipotent hematopoietic stem cells toward the B cell lineage (SOX4)[30], displayed the strongest effect in these cell types, and showed stronger activity in the early stages of each differentiation lineage. Furthermore, scooby captured early lineage commitment within G/M progenitor cells, as their TF motif effect scores closely resembled those of both differentiated myeloid cells and the progenitor populations. Additionally, the model identified repressors of gene expression such as BACH2, a TF known to be repressive of the myeloid program in B cells[31]. However, we also observed that TFs with similar motifs are scored similarly (that is, GATA1, TRPS1 and GATA3; Extended Data Fig. 7a), which is a limitation of TF binding site matching by motifs only. Despite this caveat, scooby's ability to distinguish lineage-specific patterns of TF activity suggests its potential for further exploration of the regulatory mechanisms underlying cell-fate decisions.

TFs can exert distinct effects on accessibility and expression due to temporal lags and repression mechanisms, some of which operate independently of chromatin accessibility changes. To support the investigation of motif effects on those two regulatory layers, we defined a TF motif effect score on the overall chromatin accessibility across a gene locus, analogously to the TF motif effect score on expression (Fig. 3a and Methods). Applied to GATA1, an established master regulator during erythropoiesis, in silico alteration of its binding sites indicated an early impact on chromatin accessibility across loci, whereas the effect on gene expression was delayed (Fig. 3e). This is consistent with the role of GATA1 as a pioneer factor and shows that scooby can be used to delineate the effect of motifs on accessibility from the effect on expression[32].

## scooby suggests cell-state-specific TF target genes

Identifying the genes regulated by a TF in a specific cell state is important to pinpoint the regulatory cascades driving cell-fate decision and differentiation. With scooby, we can obtain TF target genes by investigating the TF motif effect on a per-gene basis. We applied scooby to explore putative target gene regulation by three key erythroid regulators—GATA1, TAL1 and KLF1. Target genes were defined as genes predicted to show differential expression in erythroid cell types upon cognate motif mutation (Extended Data Fig. 7b and Methods). While direct validation of these predicted targets using experimental data such as from chromatin immunoprecipitation using sequencing (ChIP–seq) is beyond the scope of this study, these putative target genes were overall enriched for Gene Ontology (GO) terms related to erythropoiesis (heme biosynthetic process $P = 2.5 \times 10^{-8}$, regulation of erythrocyte differentiation $P = 5.7 \times 10^{-6}$; Methods), consistent with the known roles of GATA1, TAL1 and KLF1. As expected, the erythroid master regulator GATA1 was predicted to affect the largest number of genes. For TAL1, known to bind cooperatively with GATA1 (ref. 33), the model recapitulated the connection to known target genes such as *HBB*, *SLC4A1* and *TRIM10*, as well as other erythroid regulators (*GATA1*, *KLF1*)[33]. Finally, we observed distinct clusters of genes controlled by combinations of GATA1, TAL1 and KLF1. For instance, cluster 6 (*RHAG*, *RHD*, *ALAS2*, *TFRC*, *TSPO2*), enriched for iron ion homeostasis and ammonium transmembrane transport GO terms, was shown to be impacted by both GATA1 and KLF1 deletion. These predictions are corroborated by previous reports in human and mice (as reviewed in Perkins et al.[34]). Altogether, our analysis demonstrates that scooby can be used to investigate the complex regulatory roles of lineage-specific TFs on target genes.

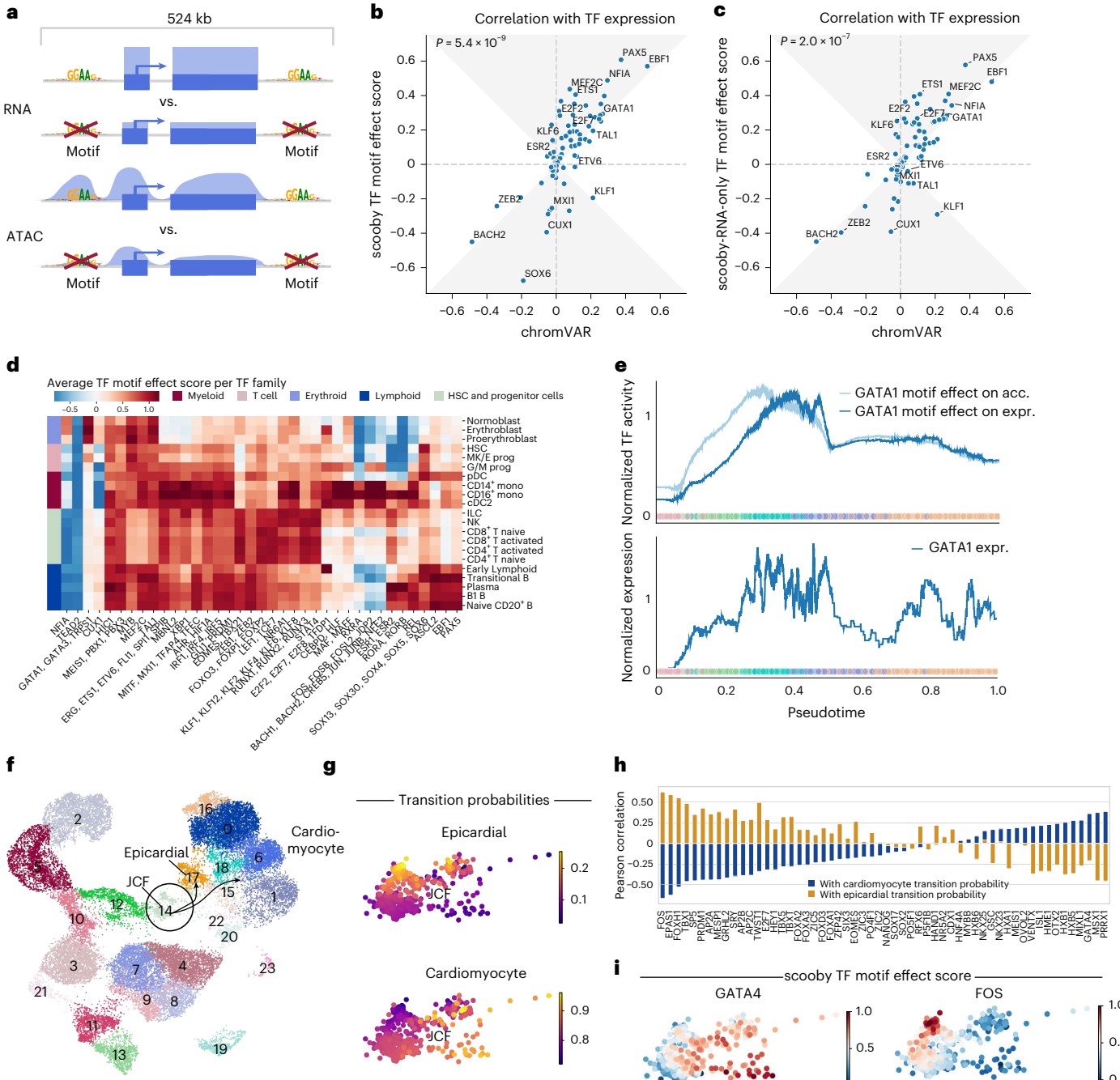

**Fig. 3 | In silico motif mutation enables TF motif effect scoring and reveals lineage and cell-state-specific regulators. a**, Schematic of TF motif effect scoring via in silico motif mutation. **b**, Pearson correlation of TF motif effect score with TF expression for scooby against chromVAR. The gray area marks the zone of improvement. We used a one-sided Wilcoxon test to compute the *P* value. **c**, Same as **b** for a scooby model trained on scRNA-seq only. **d**, Heat map of average TF motif effect score per TF family (columns) across cell types (rows). **e**, Median-normalized effect of GATA1 in silico motif mutation on accessibility and expression (top) and median-normalized expression of *GATA1* along the diffusion pseudotime axis representing erythropoietic differentiation (bottom).

Dots are colored by cell type, and lines are smoothed with a rolling mean (window size, 200 cells). **f**, UMAP visualization of multiomic metacells obtained from paired scRNA-seq and scATAC-seq data of epicardioid cells across multiple days, colored by cell type. The JCF cells (circle) and their transitions (arrows) to their two descendant cell types—cardiomyocytes and epicardial cells—are highlighted. **g**, CellRank transition probabilities toward epicardial and cardiomyocyte states within the JCF population. **h**, Correlation of TF motif effect scores with transition probability toward the cardiomyocyte (blue) and epicardial fate (yellow). **i**, Min−max scaled TF motif effect scores of GATA4 (left) and FOS (right) in the JCF cluster.

## scooby dissects TF activity within a cell type

While previous analysis focused on TF motif effects across distinct lineages, we next investigated scooby's ability to resolve TF activity within a defined cell type. To this end, we trained scooby on a published multiome dataset of human heart organoids[35], leveraging the experimentally validated heterogeneity within the juxta-cardiac field progenitors (JCFs; Fig. 3f and Extended Data Fig. 8a,b). Specifically, the original study used lineage tracing to demonstrate the dual fate of JCF progenitors, showing their differentiation into either cardiomyocytes or epicardial cells. This dual fate was mapped in scATAC-seq

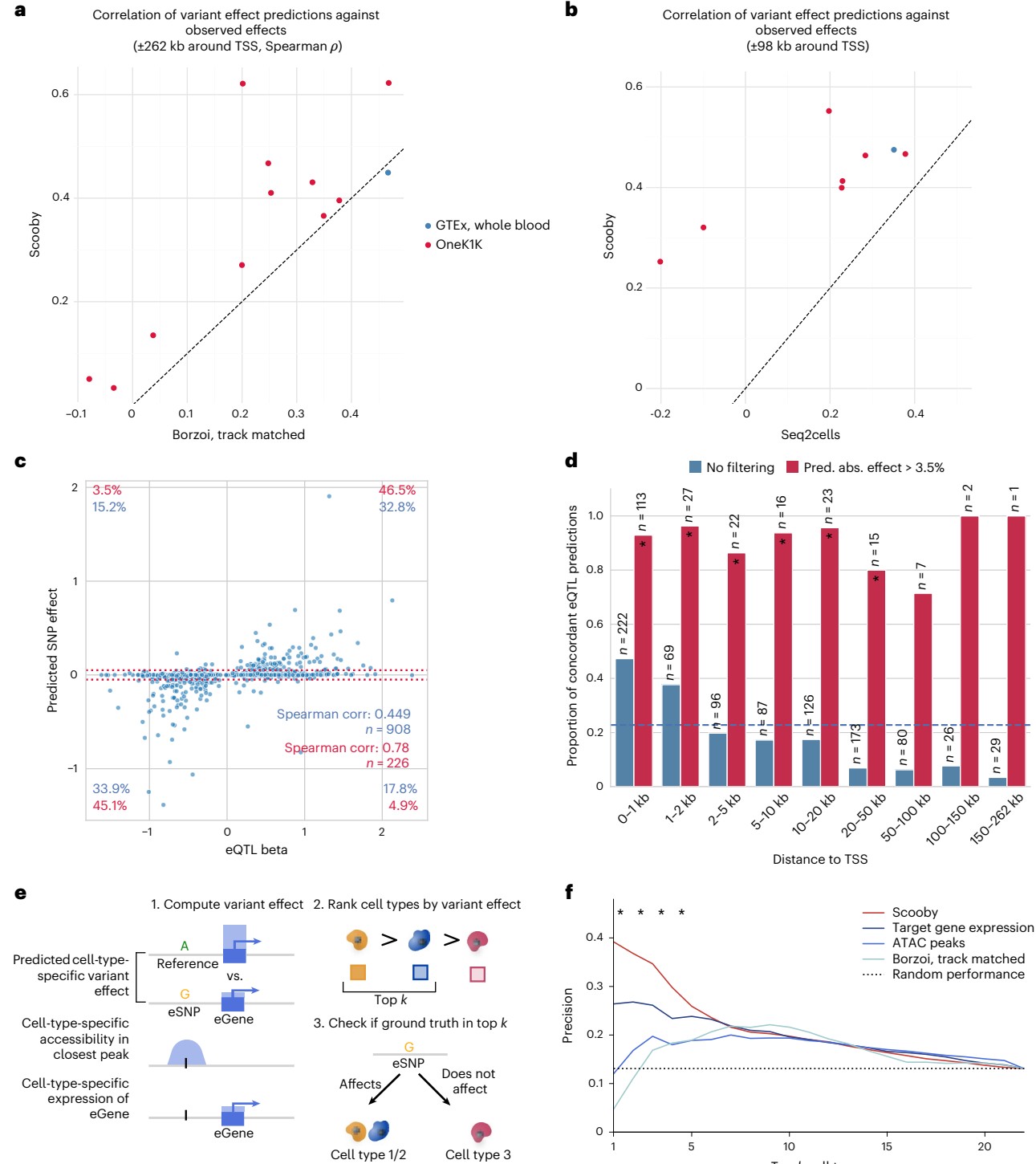

**Fig. 4 | scooby-predicted variant effects are concordant with reported effects for bulk and single-cell eQTL studies and exhibit cell-type specificity.**
**a**, Spearman correlation of predicted effects (log-fold change) with observed normalized eQTL effects for scooby against Borzoi. Each point indicates a cell type (OneK1K) or a tissue (GTEx). Dashed line marks y = x. Scooby significantly outperforms track-matched Borzoi across the OneK1K cell types (Wilcoxon rank-sum, two-sided, $P = 0.001$). **b**, Same as **a**, but for scooby against seq2cells. Scooby significantly outperforms seq2cells (Wilcoxon rank-sum, two-sided, $P = 5 \times 10^{-4}$). **c**, Predicted aggregated effects (log-fold change) versus observed whole-blood eQTL effect sizes. Red dotted lines mark thresholds below which predicted fold changes are deemed negligible (absolute fold change, 3.5%). Percentages quantify variants within each quadrant: blue indicates all variants; red denotes variants passing the 3.5% predicted effect threshold. **d**, Proportion of concordant eQTL predictions (same direction as observed), as a function of distance to the

TSS when filtering for non-negligible predicted effect (red) or without filtering (blue). Dashed blue line indicates the mean proportion of concordant eQTL predictions across all distances (0.23). Stars indicate significance over random performance (Binomial test). **e**, Schematic of the cell-type-specific evaluation. Cell-type-specific effects are obtained from model predictions, from the cell-specific accessibility in the peak closest to the variant position or via the pseudobulk expression levels of the eGene. For each approach, cell types are ranked by the absolute magnitude of the effect to distinguish cell types with and without fine-mapped eQTL associations. **f**, Precision in recovering cell types with fine-mapped eQTL associations when considering the top $k$ most highly ranked cell types using methods from **e**. Asterisks indicate significance when comparing Scooby to the target gene expression baseline (two-sided Fisher exact test, $P = 0.04, 0.02, 0.01, 0.03$). eSNP, eQTL single-nucleotide polymorphism.

and scRNA-seq data using CellRank[36,37], identifying JCF subpopulations with higher transition probabilities toward each fate (Fig. 3g). Subsequently, gene regulatory network modeling[35] was used to infer putative TFs regulating JCF lineage commitment. We leveraged scooby's TF motif effect scores computed for TFs with a differential expression of five-fold or more in at least one cell type to independently identify putative drivers of JCF lineage commitment by correlating them with CellRank-derived transition probabilities (Fig. 3h,i and Methods). We observed high correlations for bona fide epicardial fate factors, such as FOS, EPAS1, TBX1 and TFAP2A/TFAP2B/TFAP2C, and for cardiomyocyte fate regulators like GATA4, MSX1 and ISL1 (refs. 35,38,39). In summary, these results demonstrate scooby's potential to uncover single-cell heterogeneity and its ability to dissect TF activity within a cell type.

## scooby improves cell-type-specific variant effect prediction

Accurately predicting the regulatory impact of genetic variants on cell-state-specific gene expression remains a major challenge in genomics[40–42]. While sequence-based models including Borzoi and its predecessor Enformer have shown promise in distinguishing causal expression quantitative trait loci (eQTLs) from those in linkage disequilibrium[7,8], these analyses were limited to bulk data such as the tissue-specific eQTLs collected by the GTEx project[43]. Here, we leveraged the OneK1K cohort, a large-scale single-cell eQTL resource comprising over 1 million peripheral blood mononuclear cells (PBMCs) from 982 donors with statistically fine-mapped, cell-type-specific eQTLs[44,45], which we used to assess scooby's capabilities to predict cell-type-specific eQTL effects. Moreover, we used bulk whole-blood eQTLs from the GTEx project[43,45] to compare scooby to bulk RNA-seq gene expression predictors.

We trained a scooby model on the OneK1K dataset, which is derived from the same tissue as the GTEx whole-blood resource (Methods and Extended Data Fig. 8b,c). We first benchmarked scooby against Borzoi, evaluated using its corresponding GTEx whole-blood track for the GTEx eQTLs and cell-type-matched RNA-seq tracks for OneK1K eQTLs (Methods). While scooby nearly matched Borzoi's performance on GTEx whole-blood eQTLs (0.45 versus 0.47, Spearman correlation), scooby significantly outperformed Borzoi across all cell types on the OneK1K cohort (Fig. 4a). We additionally compared scooby to seq2cells, which we could only train on a subset of 100,000 cells of the OneK1K dataset due to its poor scalability. Scooby significantly improved upon seq2cells on both GTEx whole blood and the OneK1K cohort eQTLs on the common variant–gene subset (Fig. 4b).

Investigating individual whole-blood GTEx eQTL predictions, we found that scooby predicted a high proportion of variants to have negligible effects (with $\log_2$ effect less than 0.05, that is, 3.5% fold change; Fig. 4c), akin to seq2cells as originally reported by the authors[12] and recapitulated here (Extended Data Fig. 9a). Discarding these small-effect predictions, correlations on the remaining set of eQTLs increased from 0.45 to 0.78 with most predictions having the correct sign (sign concordance of 91.6%). Borzoi predictions showed the same qualitative behavior on the same set of eQTLs, indicating that this is not a characteristic of scooby itself (Extended Data Fig. 9b).

Across all predictions, the fraction of sign-concordant predictions exceeded 47% when the fine-mapped variant was located within 1 kb of the TSS and then declined with distance (Fig. 4d). This shows that, like for other sequence-based models[7,8,40], capturing the effects of distal regulatory elements on gene expression remains a challenge for scooby. Nonetheless, non-negligible predictions ($\log_2$ effect greater than 0.05) remain concordant in sign independently of the distance to the TSS (Fig. 4d and Extended Data Fig. 9c,d for seq2cells and Borzoi). These results suggest that scooby should advantageously be applied by focusing on its strong effect predictions, whereas predictions of negligible effects may not provide reliable evidence of lack of effects.

For many applications it is interesting not only to predict effect sizes but also to identify the cell type in which a variant acts. To assess the capacity of scooby at deconvolving bulk eQTLs into their cell-type-specific effects, we used the cell-type-specific fine-mapped eQTLs of OneK1K as ground truth. Specifically, among variants with a scooby-predicted variable effect across cell types, we defined the positive set to contain all cell type–variant pairs for which there was a significant fine-mapped eQTL in the OneK1K cohort. We then assessed whether scooby predicted a higher variant effect in these cell types compared to cell types without significant fine-mapping hits (Methods). We compared scooby to two realistic baseline approaches reflecting the common situation in which an expert has access to generic single-cell omics data but not to extensive individual-level single-cell omics data from genotyped cohorts like OneK1K. In the first baseline approach, cell-type specificity was inferred using the cell-type-specific accessibility in the closest or overlapping ATAC peak, leveraging accessibility data of the same cell types from a different study[46] (Methods). In the second baseline, cell-type specificity was inferred by ranking cell types according to pseudobulked expression levels of the target eGene across the entire cohort (Fig. 4e). Scooby outperformed both baseline approaches as well as the cell-type matched Borzoi (Fig. 4f). This indicates that scooby can be used to resolve cell-type-specific variant effects better than current sequence-based bulk models and simple baselines.

## scooby allows cell-type-specific delineation of bulk eQTLs

While the OneK1K eQTL analyses provided validation for scooby's ability to deconvolve eQTLs, scooby could also be used to uncover regulatory mechanisms in situations where no cell-type-specific ground truth is available. Therefore, we chose to perform a case study in which we deconvolved GTEx bulk whole-blood eQTL effects in bone marrow, using the scooby model trained on the NeurIPS bone marrow dataset.

While predicted eQTL effects generally agreed across cell types, substantial variations were observed, reflecting relationships between cell types (Fig. 5a). Notably, the erythroid, the monocyte and the early progenitor cell types each showed distinct predicted eQTL effects. To explore the potential functional relevance of this cell-type specificity, we focused on the 15% most-variable eQTLs previously associated with a human trait in genome-wide association studies (GWAS Catalog[47]; Fig. 5b and Methods). We found several cases where predicted cell-type-specific effects were consistent with the biology of the associated GWAS traits. For example, the *SLC14A1* eQTL, linked to 'Immature fraction of reticulocytes' exhibited strong effects specifically within the erythroid lineage. Similarly, the *MIR34AHG* and *NDST1* eQTLs, both associated with 'Monocyte count' showed pronounced effects in the monocytes. These findings suggest that scooby can provide insights into the cellular context of GWAS associations.

As an illustrative example of how scooby can reveal cell-type-specific regulatory mechanisms, we examined an eQTL (variant rs143664050) with a negative effect on testin (*TES*) expression in CD14[+] monocytes yet a negligible effect in erythroblasts (Fig. 5c). The alternative allele was associated with a predicted loss of an accessible region in monocytes, which could explain the observed reduction in *TES* expression. In contrast, no change in accessibility was predicted in erythroblasts, consistent with the negligible predicted effect on expression. Applying a gradient-based model interpretation method indicated that the eQTL disrupts a predicted binding site for the TF SPI1 (Fig. 5c and Methods). Notably, *SPI1* is only expressed in myeloid cells, including monocytes, but not in erythroblasts (Fig. 5d), providing an explanation for the observed cell-type specificity of this eQTL's effect (Fig. 5e). In contrast, the variant rs62032983 provided an example of an eQTL predicted to reduce expression of the gene *DCTN5* across all cell types. Model interpretation attributed this effect to the disruption of a predicted binding site for the ubiquitously expressed TF ELF1 (Extended Data Fig. 10). Altogether, these analyses demonstrate scooby's ability to link cell-type-specific variant effects, which would be missed in bulk-level analyses, to the underlying regulatory mechanism.

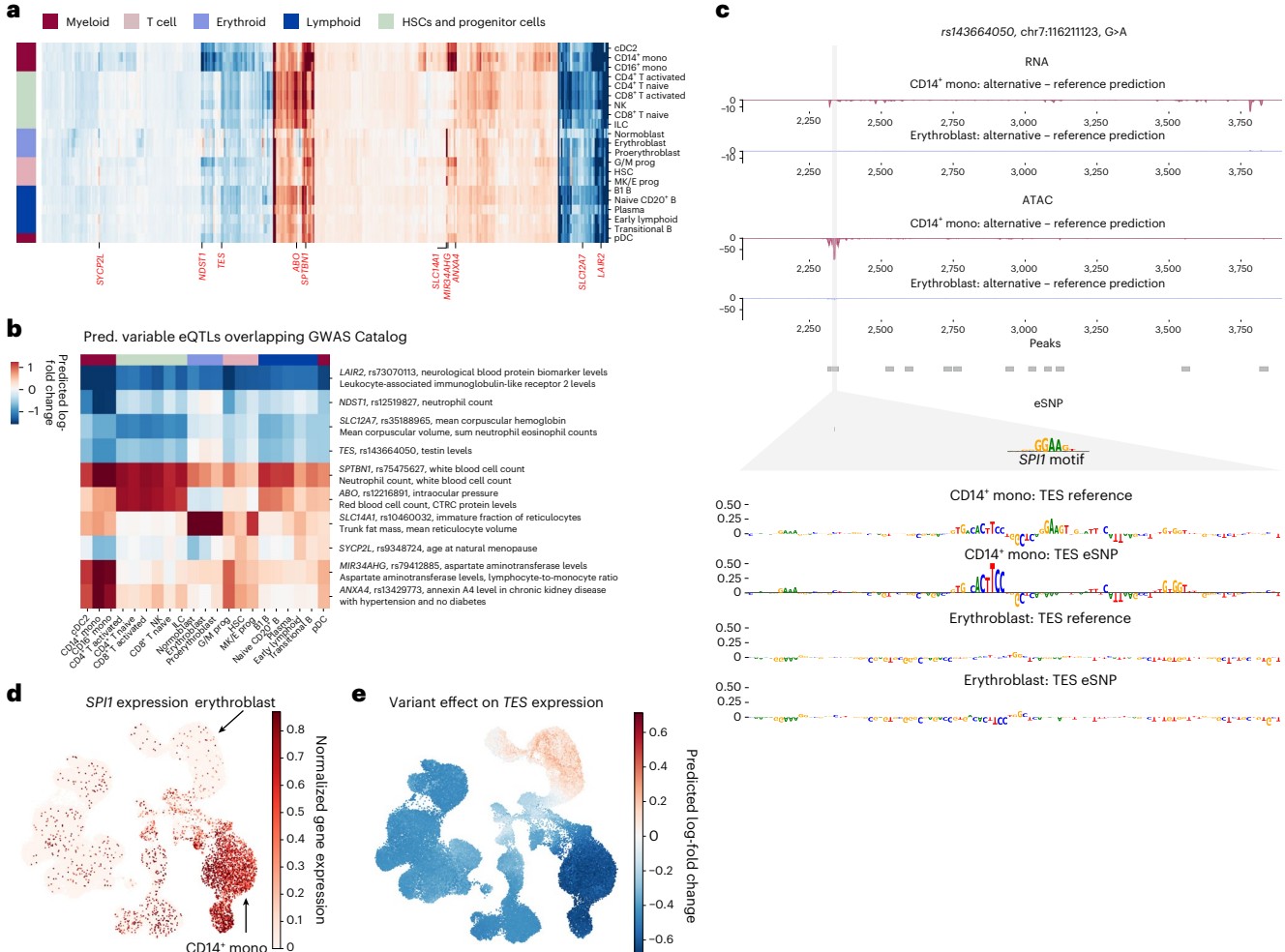

**Fig. 5 | scooby allows cell-type-specific delineation of bulk eQTLs.**
**a**, Clustermap of eQTL effect size predictions across cell types. Left color bar indicates lineage membership. Genes were clustered according to their predicted effect size per cell type. Highlighted genes (and their fine-mapped eQTLs) have predicted variable effects (black) and were considered for an overlap with the GWAS Catalog. eQTL-GWAS term matches are colored in red. **b**, Heat map of gene–variant pairs with strong cell-type-specific effects and matching GWAS

terms. **c**, Predicted fold change in gene expression (top) and accessibility (bottom) between the alternative and reference alleles of variant rs143664050 in CD14+ monocytes and erythroblasts. Sequence attributions revealed the destruction of an *SPI1* motif to only affect model outputs in CD14+ monocytes (Methods). **d**, UMAP of the NeurIPS dataset colored by observed normalized *SPI1* expression levels. **e**, UMAP showing the effect of variant rs143664050 on *TES* expression levels.

## Discussion

This work introduced scooby, which models single-cell gene expression and chromatin accessibility profiles directly from sequence contexts of half a megabase while scaling efficiently with the number of cells. This is achieved by equipping the pretrained multiomics profile predictor Borzoi with a cell-specific decoder and fine-tuning its sequence embeddings. The model shows generalizability across cells and cell types and improved the state-of-the-art model in single-cell gene expression prediction from a Pearson correlation of 0.77 to 0.87 on unseen sequences. In silico motif mutations led to TF motif effect scores showing strong concordance with TF expression levels, improving upon TF activity inference methods utilizing chromatin accessibility data. Strong concordance also held for a scooby model when training exclusively on RNA-seq data, suggesting that it can effectively leverage sequence information to infer TF motif effect scores without requiring matched accessibility data. Furthermore, we used scooby to dissect regulatory mechanisms within a seemingly homogeneous population of JCF progenitors, showcasing scooby's potential to propose hypotheses on drivers of cell-fate decisions at a finer resolution. Lastly, leveraging scooby's single-cell resolution together with interpretation

methods allowed for finer-grained analysis of variant effects, uncovering cell-type-specific eQTLs that are masked in bulk studies and the underlying TFs.

The architecture of scooby allows incorporating further modalities in two ways. Firstly, profile-based modeling is a generic approach that flexibly permits the prediction and interpretation of a wide range of additional single-cell modalities, such as methylation or ChIP–seq, in contrast to genome annotation-based methods. Secondly, scooby can in principle work with any cell-state representation. For instance, incorporating other data modalities such as CITE-seq into the embedding could allow for a finer resolution of cellular state.

We chose to mainly evaluate scooby on a 10x multiome hematopoiesis dataset, as it provides both paired scATAC-seq and scRNA-seq profiles for joint modeling and well-characterized differentiation lineages for validating TF motif effects and target gene predictions, making it an ideal test bed for our study. As scooby predicts RNA-seq coverage, it could in principle be used to predict differential isoform usage; however, the 10x scRNA-seq 3′ coverage bias limits the signal for splice sites and TSS choice[15]. Furthermore, we observed little evidence for differential isoform usage in the dataset, concordant with

studies finding that alternative transcript usage is most pronounced in brain and muscle tissues[48]. Thus, future work applying scooby to more diverse cell types or applications to alternative single-cell protocols such as SMART-seq[49] or long-read sequencing[50] is needed to assess the potential of scooby to model isoform-specific expression.

We obtained promising results when comparing scooby-based variant effect prediction with reported eQTL effects. Of practical relevance, when scooby predicted non-negligible effects for an eQTL, the effect direction was typically correct and the predicted cell-type specificity was more accurate than baseline approaches including cell-type-specific target gene expression and chromatin accessibility of the variant. This indicates that scooby can be useful to delineate cell-type specificity of eQTLs established on bulk data and to provide mechanistic hypotheses about GWAS hits. As reported previously for other sequence-based models[40], we also observe scooby to excessively underestimate the amplitudes of the effects of distal eQTLs, suggesting that this remains an area of improvement for future models.

We have demonstrated scooby's easy applicability to a large single-cell resource of 1.2 million cells[44], and to a small heterogeneous organoid dataset[35] using the same hyperparameter settings. In the future, we envision scooby to aid interpretation of large single-cell atlases[51] as the framework shows robust and efficient scaling behavior to large numbers of cells. This could be further improved by replacing the underlying base model, Borzoi, by a more efficient version[52]. Moreover, we foresee its application to learn about conserved cell-type-specific regulation by integrating diverse multispecies datasets. To facilitate adoption, we provide a streamlined workflow for applying scooby to new datasets.

In summary, scooby establishes a paradigm for connecting single-cell genomics and sequence-to-function modeling. Its modular nature and ability to integrate multimodal data and to capture cell-state-specific gene expression dynamics positions it as a valuable tool for uncovering the genetic basis of gene regulation and complex traits at single-cell resolution.

## Online content

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

## Methods

### Data acquisition and processing

**NeurIPS hematopoiesis dataset.** We obtained scRNA-seq and scATAC-seq data for the multiome hematopoiesis dataset used in the NeurIPS 2021 challenge[21]. Specifically, we downloaded scRNA-seq BAM files from the Sequence Read Archive (SRA) under accession SRP356158 and scATAC-seq fragment files from the AWS bucket s3:// openproblems-bio/public/post_competition/multiome/. Preprocessed gene count and peak count matrices were retrieved from the Gene Expression Omnibus (GEO) under accession code GSE194122.

We performed all scRNA-seq data analyses using Scanpy (v1.10)[53]. We utilized the predefined filtered cell and gene sets, as well as the highest-resolution cell-type annotations (l2_cell_type key) provided in the original publication. We identified and removed doublet cell populations using Scrublet with default parameters[54]. Doublet calls were based on a threshold that primarily captured cells clustering in discrete locations on the UMAP embedding.

We normalized raw gene expression counts using the normalize_total function in Scanpy and applied a log(expression + 1) transformation for downstream analyses. We inferred pseudotime trajectories on the integrated dataset using diffusion pseudotime[23] with default parameters.

**Epicardioids dataset.** Raw sequencing data for scATAC-seq and scRNA-seq from the original publication[35] were retrieved from the SRA with accession numbers SRP359250 and SRP359249, respectively. We aligned the scATAC-seq data using Cell Ranger ATAC (v2.1.0) with the Cell Ranger reference package refdata-cellranger-arc-GRCh38-2020-A-2.0.0. Similarly, scRNA-seq data were processed using Cell Ranger (v8.0.1) with refdata-gex-GRCh38-2020-A. In the original study, scGLUE[55] was used to generate a joint embedding, and pseudo-multiome metacells were constructed by pairing RNA and ATAC cells from this embedding. We obtained the preprocessed gene count and peak count matrices cluster labels (leiden_res1 key), scGLUE embeddings, CellRank[36,37] transition probabilities and metacell mappings from the original publication. For downstream analyses, we retained cells for which we had a scRNA-seq and scATAC-seq match.

**OneK1K dataset.** Raw sequencing data for scRNA-seq from the original publication[44] were obtained from the SRA under accession number SRP359840. scRNA-seq data were processed using the Cell Ranger pipeline (v6.1.1) with the reference package refdata-gex-GRCh38-2020-A. A preprocessed gene count matrix as well as cell-type labels were downloaded from CZ CELLxGENE[56] https://cellxgene.cziscience.com/collections/dde06e0f-ab3b-46be-96a2-a8082383c4a/. We removed the cell types 'Platelets' and 'Erythrocytes' to retain only immune cell types.

### Generation of cell embeddings

To prevent information leakage from the cell embeddings to the gene expression and accessibility prediction models, we identified and excluded genes and peaks present in the test and validation sets using pyRanges[57] (v0.0.129) before computing the embedding. We further filtered out genes and peaks present in fewer than 1% of all cells to reduce dimensionality.

For the NeurIPS dataset, we then used the MultiVI model from the scvi package (v1.1.2, https://github.com/lauradmartens/scvi-tools/tree/poissonmultivi/)[17,58] to generate a unified embedding of both scRNA-seq and scATAC-seq data. Following previous work demonstrating improved performance[16], we adapted the model to utilize raw fragment counts for scATAC-seq data, modeling these using a Poisson distribution instead of binarized counts (Poisson-MultiVI). Otherwise, we trained the model with default parameters, incorporating sample information as the batch key during data integration. This process generated an embedding vector of dimension 14 for each cell. The

100 nearest neighbors for the profile evaluations for each cell were computed using the Scanpy function sc.pp.neighbors.

For the NeurIPS RNA-only model, we reran the embedding generation using only the RNA modality by running the scVI model with n_latent set to 14.

Given the large number of donors in the OneK1K dataset, we used the scPoli model[59] from the scarches package[60] (v0.6.1) to create the embedding. The model was trained using the 'sample' key as the condition key and the 'cell_label' key for cell-type annotation, with all other parameters set to their default values.

### Efficient read coverage extraction for RNA-seq and ATAC-seq data

To generate the scRNA-seq and the scATAC-seq profiles used for training, we used an adapted version of SnapATAC2 (ref. 18; v1.0.1, https://github.com/lauradmartens/SnapATAC2/), an efficient Rust software package (rustup v1.28.1, rustc v1.85.0) initially designed for ATAC-seq data processing. SnapATAC2 achieves efficient storage of scATAC-seq data within the AnnData[19] format by recording only the start position and length of each fragment and supports out-of-memory reading.

In contrast to scATAC-seq data, scRNA-seq aligned reads contain split reads due to RNA splicing. Therefore, we modified the code of sp.pp.make_fragment_file function of SnapATAC2 to parse and store split reads as multiple entries in the AnnData, with each entry representing a contiguous fragment and its corresponding length (Extended Data Fig. 2).

### Processing of RNA BAM files and ATAC fragment files

Because the 10x BAM files included all reads (also low-quality ones), we filtered for reads that were marked as valid by the Cell Ranger pipeline (xf:i:25 flag) by implementing this filtering option to the sp.pp.make_fragment_file function (xf_filter = true). We used our modified sp.pp.make_fragment_file function, specifying the appropriate barcode and unique molecular identifier tags ('CB' and 'UB') for our data to convert the reads in the BAM file into a fragment file. We removed duplicate reads using SnapATAC2's automatic read deduplication. For processing of scRNA-seq reads, we set the is_paired argument to 'false' and both shift_left and shift_right arguments to 0.

We imported the processed fragment file into an AnnData object using sp.pp.import_data and the hg38 genome assembly. To ensure consistency with our scRNA-seq count data, we set min_num_fragments = 0 and used the whitelist argument to retain only cells present in the pre-filtered scRNA-seq AnnData object.

We imported ATAC-seq fragment files using the sp.pp.import_data function with the parameters described above. To model Tn5 insertion sites, we converted fragment locations into insertion sites by recording the fragment ends.

### Data preparation for training

We used the SnapATAC-processed coverage AnnData directly during training and created coverage tracks per cell on the fly. Following the procedure described for Borzoi[8], we first aggregated the coverage and insertions in a 32-bp window. For RNA profiles, we followed the same squashed-scale approach as Borzoi, but set clip_soft to 5, such that the fraction of soft-clipped values at the single-cell level was similar to the one of Borzoi's tracks. For ATAC profiles, we scaled the output by 0.05 to ensure they were on the same scale as the RNA coverage tracks. For faster evaluation on pseudobulks, we exported read aggregates per cell type to the bigWig format using sp.ex.export_coverage.

### Model

scooby builds upon Borzoi[8], a deep learning model for predicting RNA-seq profiles, which operates at 32-bp resolution on 524,288 bp of DNA sequence, outputting profiles for the center 6,144 bins (corresponding to 196,608 bp). We adapted a publicly available PyTorch

implementation of Borzoi (v0.0.2, https://github.com/johahi/ borzoi-pytorch/), removing the original human and mouse-specific output heads and retaining the convolutional and transformer layers responsible for sequence encoding. We integrated LoRA modules[14], each with a rank of 8, into all convolutional layers and the query, value and MLP projection matrices within the transformer layers using an adapted version of peft[61] (v0.10.1, https://github.com/lauradmartens/ peft/). While used separately during training, these LoRA modules were merged back into the original model weights after training, resulting in no additional overhead during inference.

Furthermore, we introduced a trainable layer with Gaussian error linear unit nonlinearity on top of Borzoi's penultimate layer. The weights of this layer were randomly initialized and trained from scratch, allowing for potential refinement of Borzoi's embeddings for the single-cell context. The output of this layer was then passed to the cell-state-specific decoder. This decoder operates on the sequence embeddings of the center 6,144 bins. Specifically, the decoder consists of a $1 \times 1$ convolution along the sequence dimension (effectively a position-wise linear transformation). The weights of this convolutional filter were not fixed but were dynamically generated for each cell based on its corresponding cell embedding. To this end, a small multilayer perceptron was added that receives the cell embedding as input and outputs a vector to parameterize the convolutional filter used to produce the final predicted profiles for that cell from the sequence embedding. We visualize the exact model architecture in Extended Data Fig. 1. For stranded RNA predictions, the multilayer perceptron outputs a weight matrix of shape (1921, 2, 1), encompassing the filter weights (1,920 dimensions) and biases (1 dimension) for each strand. For ATAC predictions (unstranded), it outputs a weight matrix of shape (1921, 1, 1).

To efficiently scale scooby to a large number of cells, we implemented two optimizations. First, we introduced a caching mechanism for the sequence embeddings, reducing redundant computations when predicting profiles for multiple cells from the same genomic region. Second, we performed cell-state-specific decoding for expression only on the embedding slices that overlap with exons (or gene body) of interest.

### Training procedure

We initialized scooby's Borzoi backbone with pretrained weights from Borzoi's replicate 0 (test fold 3, validation fold 4), converted from the original TensorFlow implementation to PyTorch. These pretrained weights correspond to a model trained on the human and mouse reference genome (hg38, mm10 assembly). We maintained the same train-val-test split as Borzoi for scooby's training. During training, we only updated the parameters of the LoRA modules, the cell-state-specific convolutional filter weights and the weights of the additional layer with Gaussian error linear unit nonlinearity introduced after Borzoi's penultimate layer. With PyTorch (v2.1.0), we used the AdamW optimizer with a learning rate of $4 \times 10^{-4}$ for the cell-state convolutional layer and $2 \times 10^{-4}$ for the LoRA modules and the additional layer to stabilize training. Both learning rates were warmed up over the first 1,000 steps and decayed linearly afterward over 40 epochs. To stabilize training during the first step, we froze the batch normalization layers from the pretrained Borzoi model and disabled dropout within Borzoi. We then unfroze the batch normalization layers and enabled dropout to prevent overfitting.

We monitored validation performance using the Pearson correlation between the predicted and observed total counts (gene count evaluation, $\log_2$-transformed pseudobulk counts with an added pseudocount of 1) and retained the model with the largest correlation across cell types.

All models were trained with a batch size of eight sequences across eight A40 GPUs, using mixed precision to accelerate training. Per training sequence, the model predicted RNA and ATAC profiles

for 64 randomly sampled cells in a multi-task learning fashion. We randomly augmented training sequences by shifting them by up to three base pairs in either direction and reverse-complementing them. To ensure consistent strand orientation, we reverse-flipped the target profiles when training on reverse-complemented sequences. We used the same weighting scheme for the Poisson and multinomial loss terms as in the original Borzoi implementation. The gradient clipping threshold was set to 1.0, and weight decay was set to $10^{-6}$ for all trainable parameters. Due to computational constraints, extensive hyperparameter optimization was not performed.

### Ablations and other models

To train the model without normoblasts, we used the same hyperparameters and training procedure as described above. However, to prevent leakage, we recreated the MultiVI embedding without normoblasts and normoblast cells were excluded from the random sampling of cells during training. To train the RNA-only ablation, we followed the above steps, but instead removed scATAC-seq targets and output heads and used an embedding based on RNA-seq only. For the model without LoRA, we removed the trainable LoRA weights while maintaining the model architecture elsewhere. We additionally trained models with scooby heads on the outputs of Borzoi when using all 7,611 output tracks, or when subsetting on the RNA-seq track only ($n = 1,543$). In contrast to all other models, these training runs diverged, which was circumvented using a lower learning rate ($5 \times 10^{-5}$).

We downloaded seq2cells from the official GitHub repository (https://github.com/GSK-AI/seq2cells)[12] and processed files to fit the required format. For comparability, we used a cell × gene matrix with counts generated by summing over the observed profile. We matched gene IDs to gene names, retaining 15,892 genes, and followed the same data split as Enformer[7], upon which seq2cells is based. We trained the model using the training configuration provided in the repository, but longer for up to 40 epochs to stay comparable to scooby, and evaluated the best checkpoint.

### Inference

To obtain model predictions, we performed inference on both the input sequence and its reverse complement. The output tracks for the reverse complement were reverse-flipped, and both predictions were averaged to produce the final profiles. We then reversed the squashed-scale transformation to obtain raw expression values, and scaled accessibility profiles by 20 to reverse the transformation applied during training. We used mixed-precision inference to accelerate computations.

### Profile-level evaluation

To quantitatively assess profile prediction accuracy, we calculated the Pearson correlation on a logarithmic scale between the predicted profiles and the observed profiles (with and without averaging across 100 cell neighbors) over all test sequences for a random subset of cells of each cell type. Additionally, we compared each single-cell profile to the pseudobulk (averaged) profile of the same cell type, and to its averaged 100 nearest-neighbor profile in the same fashion. Genomic annotations were plotted using trackplot (v0.4.0)[62].

### Gene count evaluation

To generate cell-type-specific counts, we first centered the input sequence on the gene body as annotated with GENCODE release (v32)[63]. We summed the expression profile for each bin overlapping an exon using the output track that matched the strand of the gene. This was repeated for each cell and summed across all cells of the same cell type to obtain pseudobulk counts. Finally, we $\log_2$-transformed both the predicted and target pseudobulk counts, and added a pseudocount of 1 to calculate the Pearson correlation across genes for each cell type. To obtain a metric quantifying how well the model captures

cell-type-specific expression, we subtracted the mean across genes of the cell-type × gene log$_2$-transformed expression matrix, and then subtracted the mean across cells for both predictions and observations and correlated the results. For the no-normoblast ablation, we predicted all test gene counts pseudobulked for each cell type as described above and correlated their predicted expression with the observation of the normoblast cell type. For the comparison with seq2cells, we only retained genes overlapping the Enformer test set (used in seq2cells) and the Borzoi test set (used in this study). Predicted and observed gene counts along the diffusion pseudotime axis were smoothed using a rolling window of 200 cells with mean aggregation.

### Motif deletion experiments for the NeurIPS hematopoiesis dataset

To investigate the impact of TF binding sites on scooby's predictions, we performed in silico motif deletion experiments. We first obtained a list of TF PWMs from the HOCOMOCO v12 core database (https://hocomoco12.autosome.org/final_bundle/hocomoco12/H12CORE/formatted_motifs/H12CORE_meme_format.meme)[24]. To focus on TFs with potential regulatory roles in the relevant cell types, we filtered the list for TFs overlapping the set of differentially expressed genes per lineage. Differential gene expression analysis was conducted for each lineage against the others using the Wilcoxon rank-sum test, with a significance threshold of $P \leq 0.05$ after correcting for multiple testing with the Benjamini–Hochberg procedure using the Scanpy function sc.tl.rank_gene_groups with groupby = 'l1_cell_type'. We selected the motif with the most evidence and lowest motif subtype for each TF, drawing randomly if multiple candidates exist.

We used tangermeme FIMO (v0.2.3)[64] to scan input sequences centered on differentially expressed genes, identifying putative TF binding sites based on their PWMs using default significance cutoffs of 0.0001 after converting pwm-matching log-odds scores into $P$ values. We generated alternative sequences by substituting each predicted binding site with a random nucleotide sequence of the same length, repeating this procedure ten times per sequence to mitigate spurious motif introduction.

We used scooby to predict gene expression and accessibility profiles for both the original and motif-deleted sequences in each cell. For gene expression, we summed the predicted RNA-seq coverage over all exons within a gene. For accessibility, we summed the predicted scores across the entire 6,144 × 32-bp output bins. We averaged over the ten distinct random replacements. The prediction for each cell was divided by the size factor of its corresponding reference prediction and scaled by its median reference size factor. For each cell, we calculated the mean log$_2$-fold change between the reference and alternative sequence predictions across differentially expressed genes, yielding a single TF score per cell.

For the chromVAR[25] comparison, we used pychromVAR (v0.0.4) with default configuration, except that we used the PWMs of the HOCOMOVO v12 core database for consistency. For scBasset[9], we followed the scVI tutorial (https://docs.scvi-tools.org/en/stable/tutorials/notebooks/atac/scbasset.html) to obtain TF activity scores.

To compare TF scores and target TF expression, we then computed the Pearson correlation of each TF score (from scooby, chromVAR and scBasset) with log-normalized TF expression.

To identify putative target genes of GATA1, TAL1 and KLF1 in the erythrocyte lineage (MK/E progenitor, proerythroblast, erythroblast), we examined TF motif effects on a per-gene basis, rather than gene-averaged effects. Target genes were defined as those exhibiting an absolute predicted effect size exceeding 0.1. These target genes were then clustered based on their effect sizes using seaborn's clustermap function, using the 'seuclidean' metric and 'ward' linkage method. Clusters were determined via hierarchical clustering with the SciPy (v1.13.1) fcluster function, using the criterion = 'maxclust' parameter. Gene-set enrichment analysis was subsequently performed using

Enrichr[65] through the GSEApy Python package[66] (v1.1.3), utilizing the GO Biological Process 2021 gene set.

### Motif analysis for the epicardioid dataset

The motif analysis was performed as described above except that differential expression was calculated between cell types and not lineages. Differentially expressed genes were defined using a log-fold-change threshold of 2 and an adjusted $P$-value threshold of 0.01. Moreover, for computational reasons, we restricted the analysis to the 58 differentially expressed TFs in cardiomyocytes and epicardial clusters whose maximum absolute fold change was larger than 5. The inferred TF motif effect scores were then correlated within the JCF population with the CellRank transition probabilities from the original study.

### eQTL effect benchmark

We evaluated the ability of scooby to pinpoint likely causal nucleotide variants driving gene expression changes within fine-mapped eQTLs from GTEx[43] and OneK1K. We used the publicly available summary statistics and uniformly generated fine-mapping results from the eQTL Catalog (https://www.ebi.ac.uk/eqtl/)[45]. Following Linder et al.[8], we used single-nucleotide variants with a posterior inclusion probability (PIP) $\geq 0.9$ and filtered out all non-single-nucleotide variants such as indels and deletions.

For each variant, we centered the input sequence on the eSNP and recorded the effect on gene expression (sum over exons) and accessibility (sum over the whole region) for the reference and the alternative nucleotide of the true eSNP. We then computed cell-type-level variant effects by summing the predicted gene expression levels on the natural scale over all cells of that cell type and computing the log$_2$-fold change of alternative versus reference predictions, adding a pseudocount of 1. For a general variant effect, we averaged the effect over all cell types.

To be able to compare Borzoi to scooby, we used the 'GTEX: RNA blood' track for comparisons on the GTEX whole-blood tissue, and matched the Borzoi RNA tracks to OneK1K cell types where possible, manually selecting the closest group of cell types if no matching track was found (Supplementary Table 1). For comparison to seq2cells, we trained a seq2cells model on a subset of the OneK1K dataset consisting of 100,000 randomly sampled cells of the full dataset, as a model on the full dataset used too much memory to be trainable.

### eQTL deconvolution benchmark

Because the GTEx whole-blood sample size is larger than the OneK1K cohort, we extended our OneK1K eQTL set (cell-type-level PIP $\geq 0.9$) by variants that were fine-mapped with PIP $\geq 0.9$ in GTEx blood but were below the 0.9 PIP threshold for a specific cell type in OneK1K. For each variant, we recorded the cell types the variant was fine-mapped in as our ground-truth positives, regarding all other cell types as negatives.

We then filtered for variants where scooby predicted a non-negligible effect (predicted effect size greater than 3.5%) and validated whether scooby ranks cell types belonging to cell types from the above set most highly. We used the cell-type matched Borzoi tracks to similarly rank cell types for each variant.

For the target gene expression baseline, we ranked cell types by the pseudobulked gene expression of the eGene after size-factor normalizing the single-cell counts.

For the ATAC baseline, we used a PBMC scATAC-seq dataset of 9,030 cells from 10x Genomics, which was downloaded from the Azimuth web application[46] (https://app.azimuth.hubmapconsortium.org/app/human-pbmc-atac) and contains the same cell types as the OneK1K study. We pseudobulked ATAC fragment counts per peak after size-factor normalizing each cell. We ranked cell types according to the accessibility of the eSNP's overlapping or nearest ATAC-seq peak.

### eQTL deconvolution in the NeurIPS dataset

To identify eQTLs with cell-type-specific effects, we focused on variants demonstrating both a strong overall effect and substantial variability

across cell types. We first filtered for variants with a substantial average effect size (log$_2$(mean effect) > 0.05) across all cell types. We labeled eQTLs as cell-type specific if their effect sizes showed high variability across cell types, specifically within the top 15% of variance. We clustered the variants on their effect size using seaborn clustermap with metric = 'seuclidean' and method = 'ward'. We downloaded v1.0 of the GWAS Catalog from https://www.ebi.ac.uk/gwas/docs/file-downloads and matched GTEx variants using rs_id_dbSNP151_GRCh38p7, manually adding matches where possible using dbSNP (https://www.ncbi.nlm.nih.gov/snp/)[67]. We report up to three randomly selected terms matching variant and target gene combination.

To link eSNPs to TF motifs, we generated gradient-weighted PWMs encompassing a 10-bp window centered on each variant. Specifically, we derived the PWM of the 10-bp window by performing exponentiation of 2 with the gradient of each nucleotide per position, and subsequently sum-normalized each position. We then used Tomtom (v5.5.2) to scan these PWMs against the HOCOMOCO v12 core database using the default MEME parameters[68]. To account for potential redundancy among TFs with similar motifs, we filtered the results to include only TFs expressed in at least 1% of all cells. Since TF nomenclature can vary between databases, we used a Python interface to UniProt (Unipressed v1.3.0) to map protein names from HOCOMOCO to their corresponding gene symbols.

### Writing
A large language model was used to assist with refining the phrasing and clarity of the manuscript. All suggestions generated by the large language model were carefully reviewed and edited by the authors.

### Reporting summary
Further information on research design is available in the Nature Portfolio Reporting Summary linked to this article.

### Data availability
The scRNA-seq, scATAC-seq and preprocessed count matrices for the multiome hematopoiesis dataset are available from the NeurIPS 2021 challenge, SRA (accession SRP356158), AWS (s3://openproblems-bio/public/post_competition/multiome/) and GEO (accession GSE194122). The epicardioids dataset raw data (scATAC-seq, scRNA-seq) are available from SRA (accessions SRP359250 and SRP359249). The OneK1K dataset raw data (scRNA-seq) are available from SRA (accession SRP359840). Preprocessed OneK1K data are available from CZ CELLxGENE (https://cellxgene.cziscience.com/collections/dde06e0f-ab3b-46be-96a2-a8082383c4a1/). We used the Cell Ranger references refdata-cellranger-arc-GRCh38-2020-A-2.0.0 and refdata-gex-GRCh38-2020-A. We used the GENCODE release v32 GTF file and the GO Biological Process 2021 gene set. TF position weight matrices were obtained from HOCOMOCO v12 (https://hocomoco12.autosome.org/final_bundle/hocomoco12/H12CORE/formatted_motifs/H12CORE_meme_format.meme). GTEx eQTL and OneK1K summary statistics and fine-mapping results are available at https://www.ebi.ac.uk/eqtl/. The scATAC-seq PBMC dataset was downloaded from https://app.azimuth.hubmapconsortium.org/app/human-pbmc-atac/. The GWAS Catalog (v1.0) was downloaded from https://www.ebi.ac.uk/gwas/docs/file-downloads. We matched SNP IDs using dbSNP (rs_id_dbSNP151_GRCh38p7, https://www.ncbi.nlm.nih.gov/snp/).

### Code availability
The scooby model including training scripts and data loaders are available at https://github.com/gagneurlab/scooby/. Jupyter notebooks and scripts to reproduce our analysis and figures are available at https://github.com/gagneurlab/scooby_reproducibility/. The adapted version of SnapATAC2 is available at https://github.com/lauradmartens/SnapATAC2/. The code along with data to reproduce the findings have

additionally been archived and are available via Zenodo at https://doi.org/10.5281/zenodo.15517764 (ref. 69) and https://doi.org/10.5281/zenodo.15517072 (ref. 70).

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

### Acknowledgements
We thank J. Engelmann for insightful discussions regarding variant effect prediction, M. Ozols for providing processed OneK1K data, A. Moretti and M. Gander for insights on the epicardioids data, R. Zhang for computational advice as well as T. Mauermeier for his talented cluster support. We thank P. Tomaz da Silva and N. Wagner for comments on the manuscript. L.D.M. acknowledges support by the Helmholtz Association under the joint research school Munich School for Data Science. The work was supported by funding from the Deutsche Forschungsgemeinschaft (DFG; 403584255, TRR267

to L.D.M. and J.G.), the European Research Council (101118521, EPIC to J.C.H. and J.G. and 101054957 to F.J.T.), the Gene Regulation Observatory at the Broad Institute of MIT & Harvard (to J.D.B.), the National Institutes of Health (NIH; 5P01HL131477 to J.D.B) the NHGRI IGVF consortium (UM1 HG011986 to J.D.B.) and the NIH New Innovator Award (DP2 HL151353 to J.D.B.). The views and opinions expressed are those of the authors and do not necessarily reflect those of the European Union or the European Research Council. Neither the European Union nor the granting authority can be held responsible for them. This study was supported by the Deutsche Forschungsgemeinschaft via the IT Infrastructure for Computational Molecular Medicine (461264291, 553375143).

## Author contributions

L.D.M. and J.C.H. conceived the project with J.G. J.C.H. designed the model. L.D.M. and J.C.H. implemented the framework. L.D.M. and T.M. developed the data pipeline. L.D.M. and J.C.H. designed the evaluations and analyzed the data with help from A.K. and J.D.B. J.G., J.D.B. and F.J.T. supervised the project. J.C.H., L.D.M. and J.G. wrote the manuscript with input from all authors.

## Funding

## Competing interests

J.D.B. holds patents related to ATAC-seq and is an SAB member of Camp4 and seqWell. F.J.T. consults for Immunai, Singularity Bio, CytoReason and Omniscope, and has ownership interest in Dermagnostix and Cellarity. The other authors declare no competing interests.

## Additional information

**Extended data** is available for this paper at https://doi.org/10.1038/s41592-025-02854-5.

**Correspondence and requests for materials** should be addressed to Julien Gagneur.

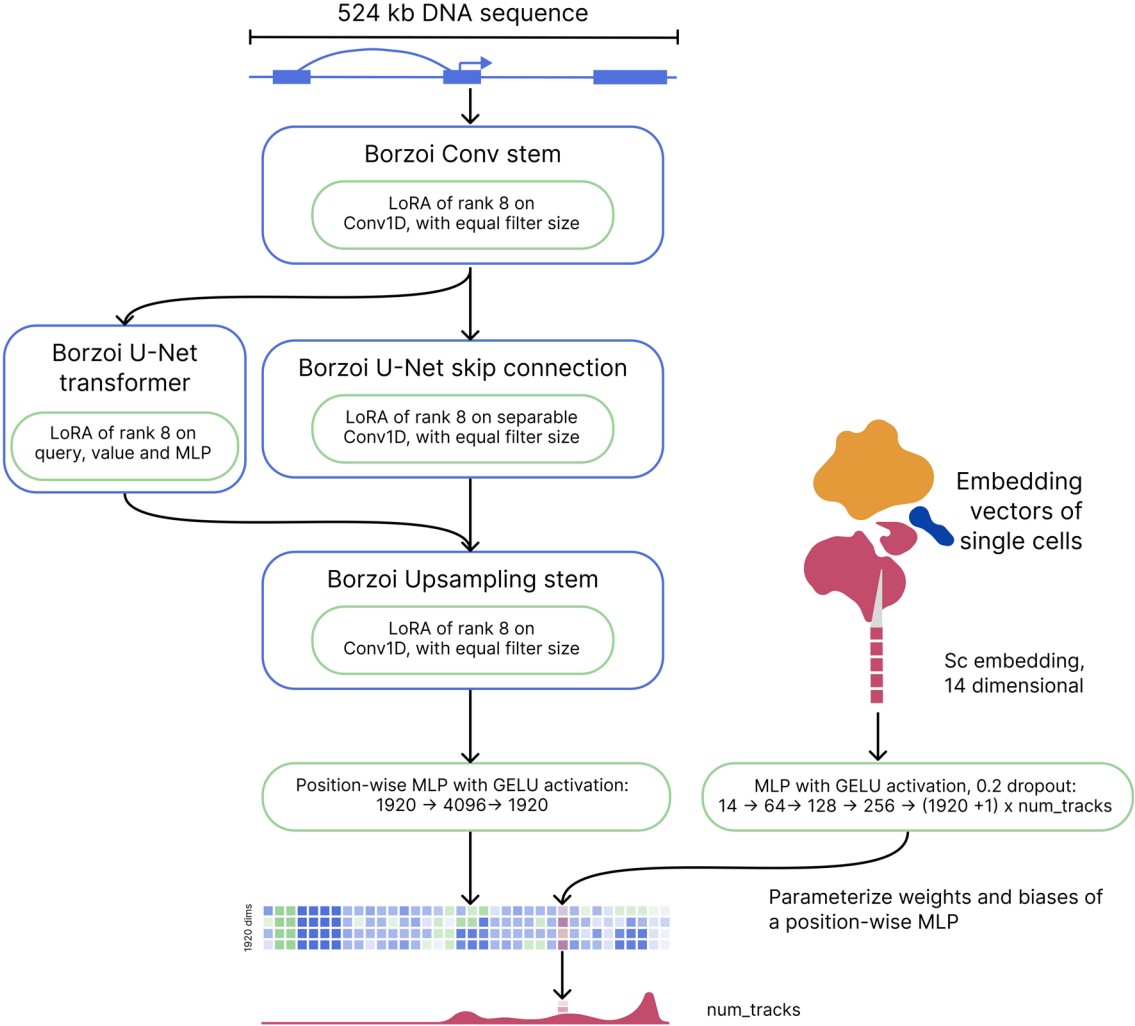

**Extended Data Fig. 1 | Scooby architecture overview.** Neural network diagram. Green boxes denote trainable parameters, blue boxes depict frozen (non-trained) model parts. 524 kb of DNA sequence is processed by a LoRA augmented Borzoi stem with an additional MLP on the Borzoi embedding (left), whereas the single cell embedding is passed through a MLP (right) to predict the filter weights (red boxes) used to decode the sequence embedding.

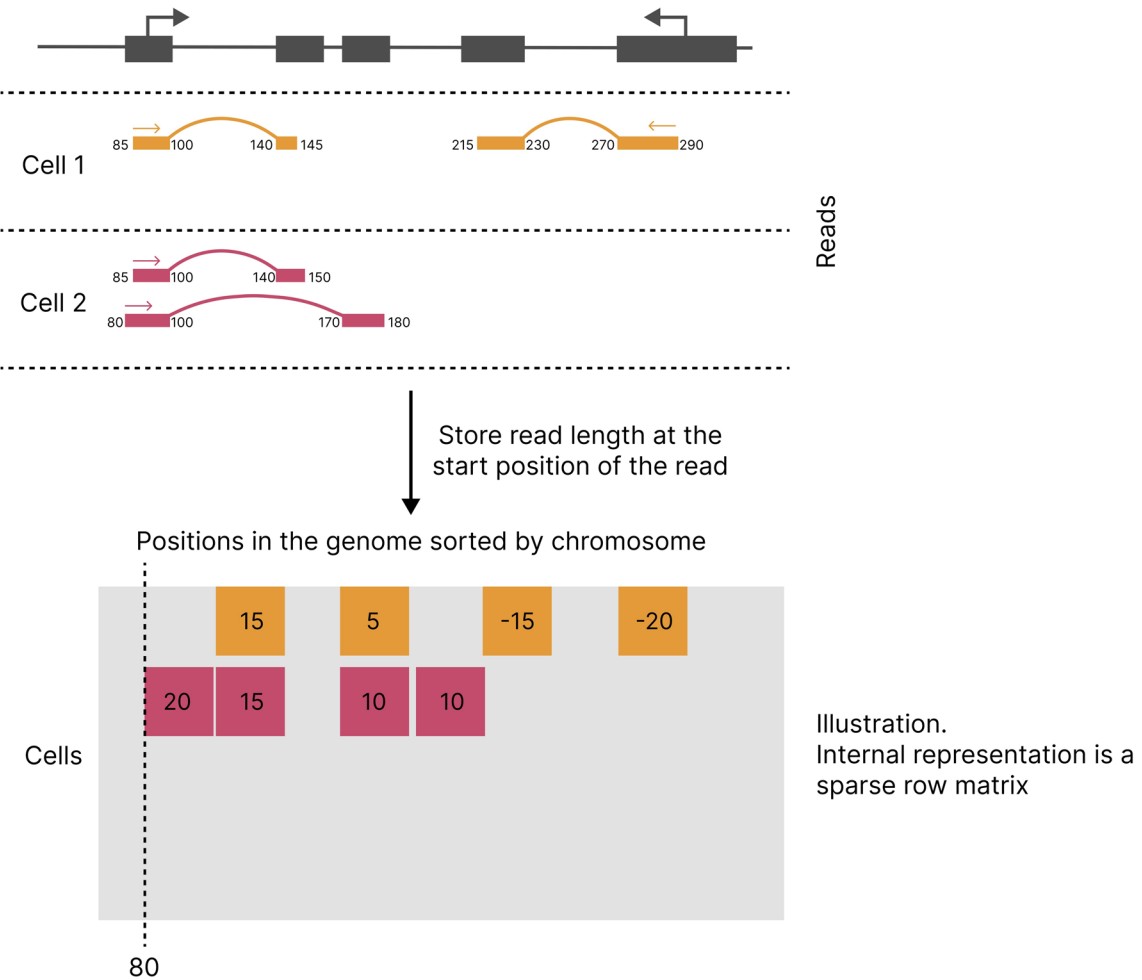

**Extended Data Fig. 2 | Memory-efficient storage of single-cell RNA-seq data.** Memory-efficient storage of single-cell RNA-seq data using a modified snapATAC2.0 format. Each row in the resulting sparse matrix represents a cell, and each column represents a genomic position. Reads are stored at their start positions with values indicating read length. Split reads are stored as multiple entries. Negative values indicate reads mapped to the negative strand.

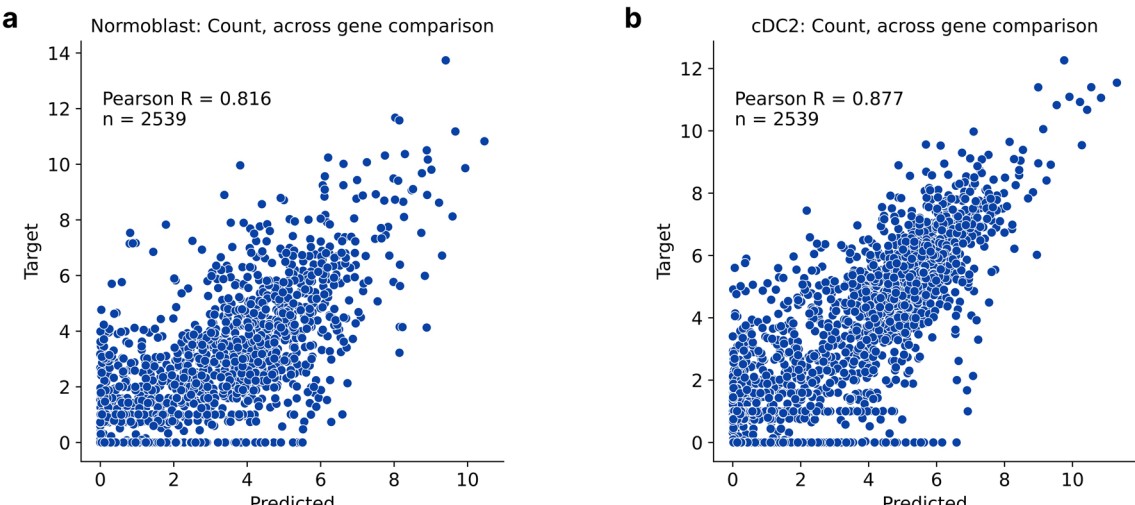

**Extended Data Fig. 3 | Scooby predicted against observed counts for two representative cell types.** Log-transformed predicted versus observed counts of scRNA-seq reads overlapping exons for the worst (**a**) and best (**b**) predicted cell type.

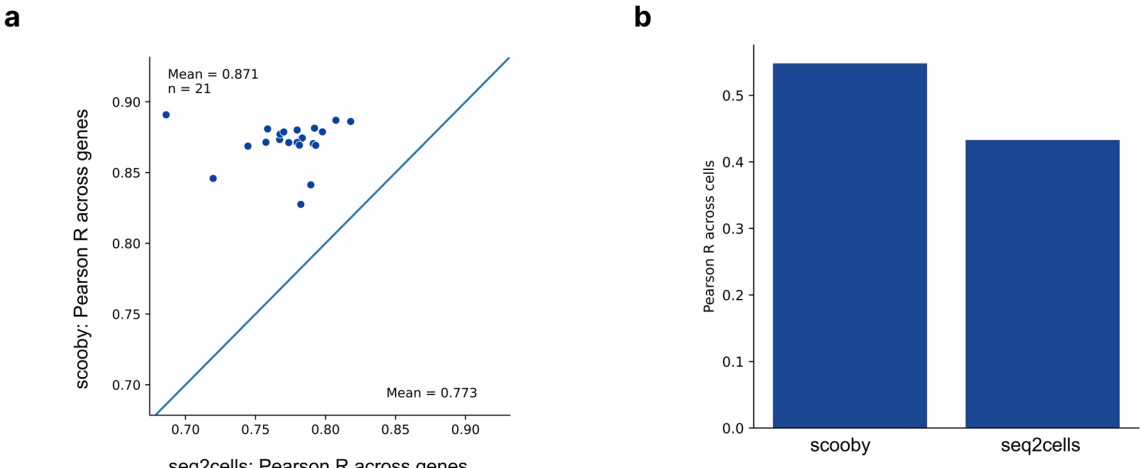

**Extended Data Fig. 4 | Scooby predicts gene expression more accurately than seq2cells. a**, Across-gene Pearson correlation for all cell types comparing scooby and seq2cells. **b**, Between-cell-type Pearson correlation after subtracting gene and cell mean gene expression comparing scooby and seq2cells.

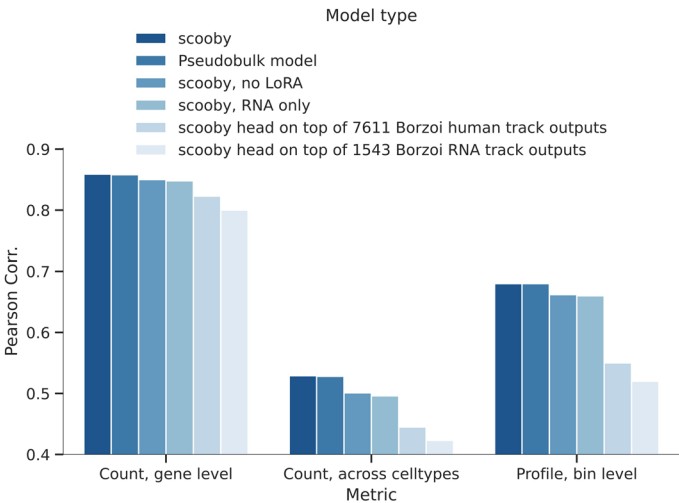

**Extended Data Fig. 5 | Scooby model ablation studies.** Performance comparison of alternative modeling approaches at predicting gene expression counts and binned profiles.

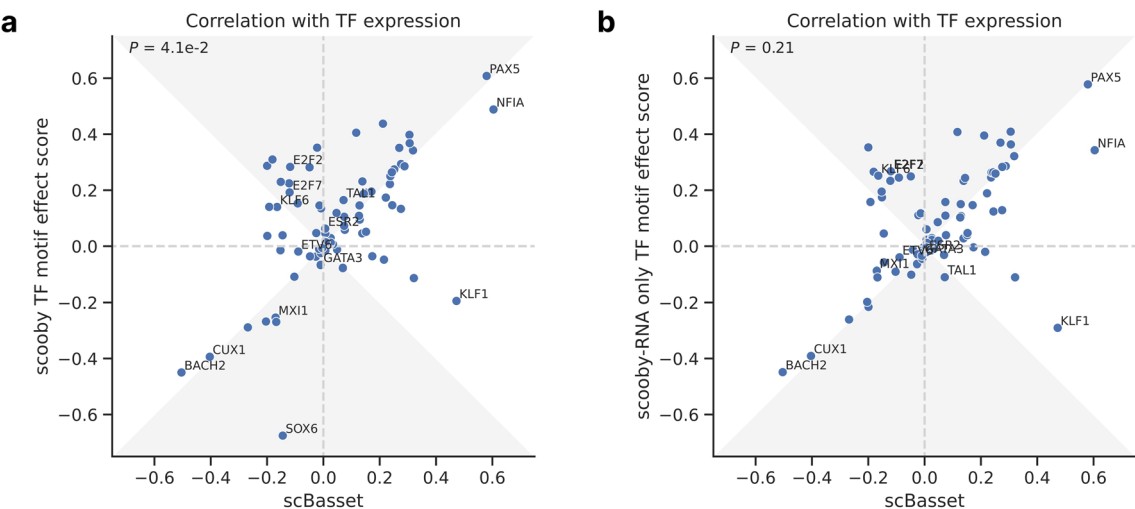

**Extended Data Fig. 6 | TF motif effect score comparison with scBasset. a**, Pearson correlation of TF motif effect score with TF expression for scooby against scBasset. The gray area marks the zone of improvement. **b**, Same as **a** for a scooby trained on scRNA-seq only.

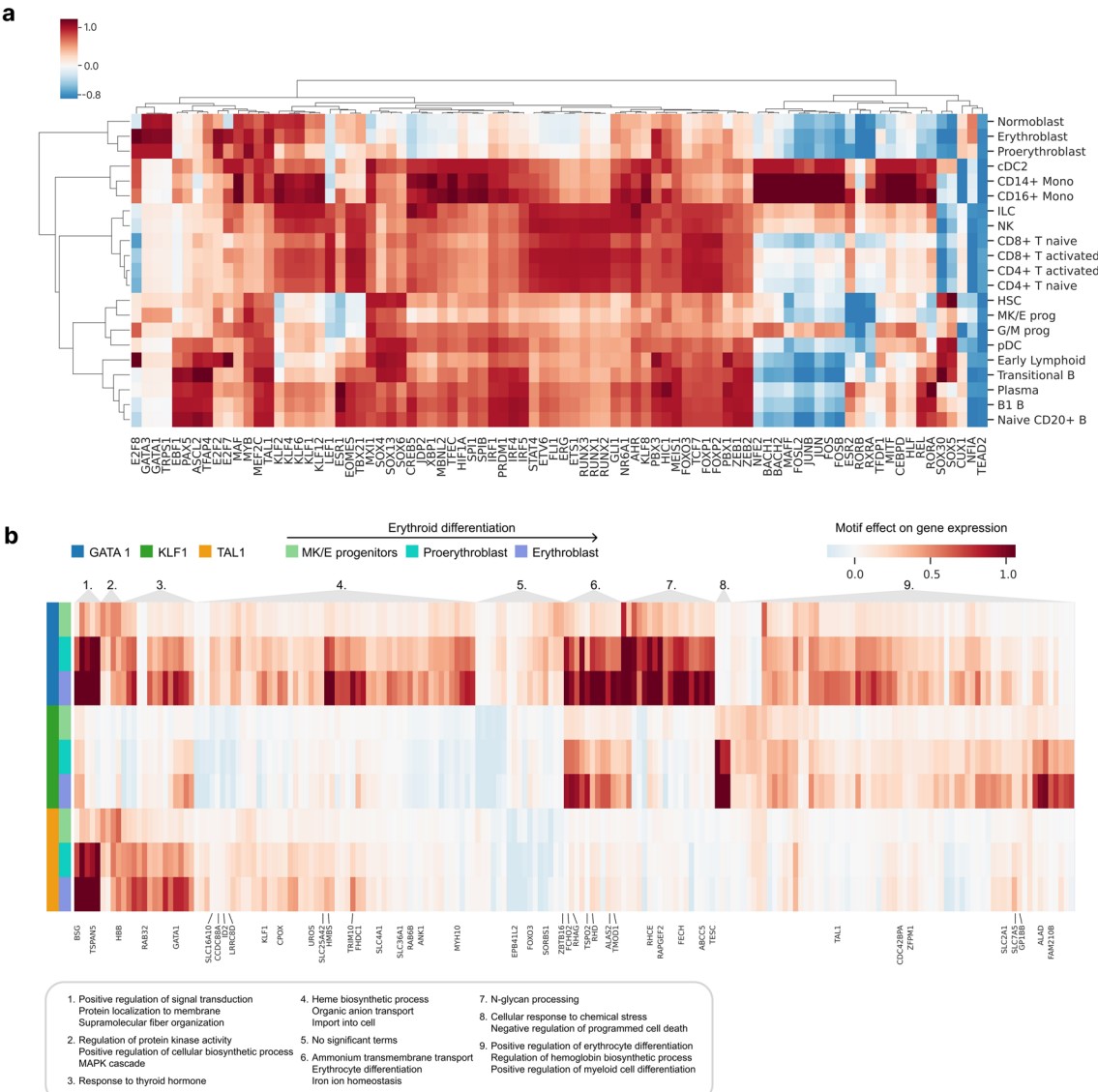

**Extended Data Fig. 7 | TF motif effect scoring allows the investigation of TFs and target gene regulation. a**, Heatmap of TF motif effect scores for differentially expressed TFs for all cell types. Genes and cell types are clustered according to their TF motif effect scores. **b**, Heatmap of genes with high motif mutation effects of GATA1, KLF1 and TAL1 in the erythrocyte lineage. Genes are clustered according to their motif effect score. The three most significantly enriched GO terms for each cluster are shown. Gene names are only shown for genes ascribed to these terms.

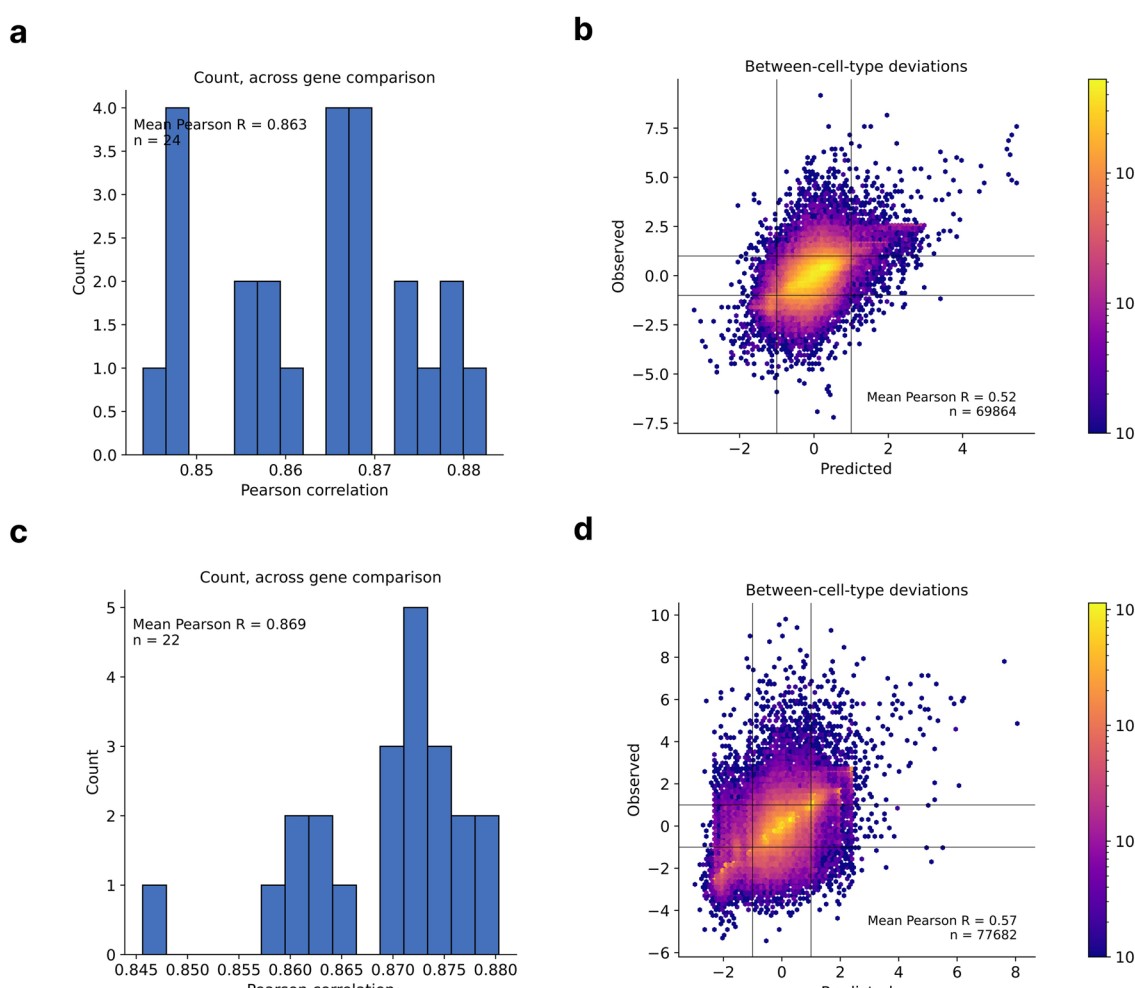

**Extended Data Fig. 8 | Scooby test set performance for the Epicardioids and OneK1K datasets.** Distribution of gene-level Pearson correlation between log-transformed predicted and observed counts of scRNA-seq reads overlapping exons across cell types for the Epicardiods dataset (**a**) and the OneK1K dataset (**c**). Predicted against measured between-cell-type deviations of gene expression for the Epicardiods dataset (**b**) and the OneK1K dataset (**d**).

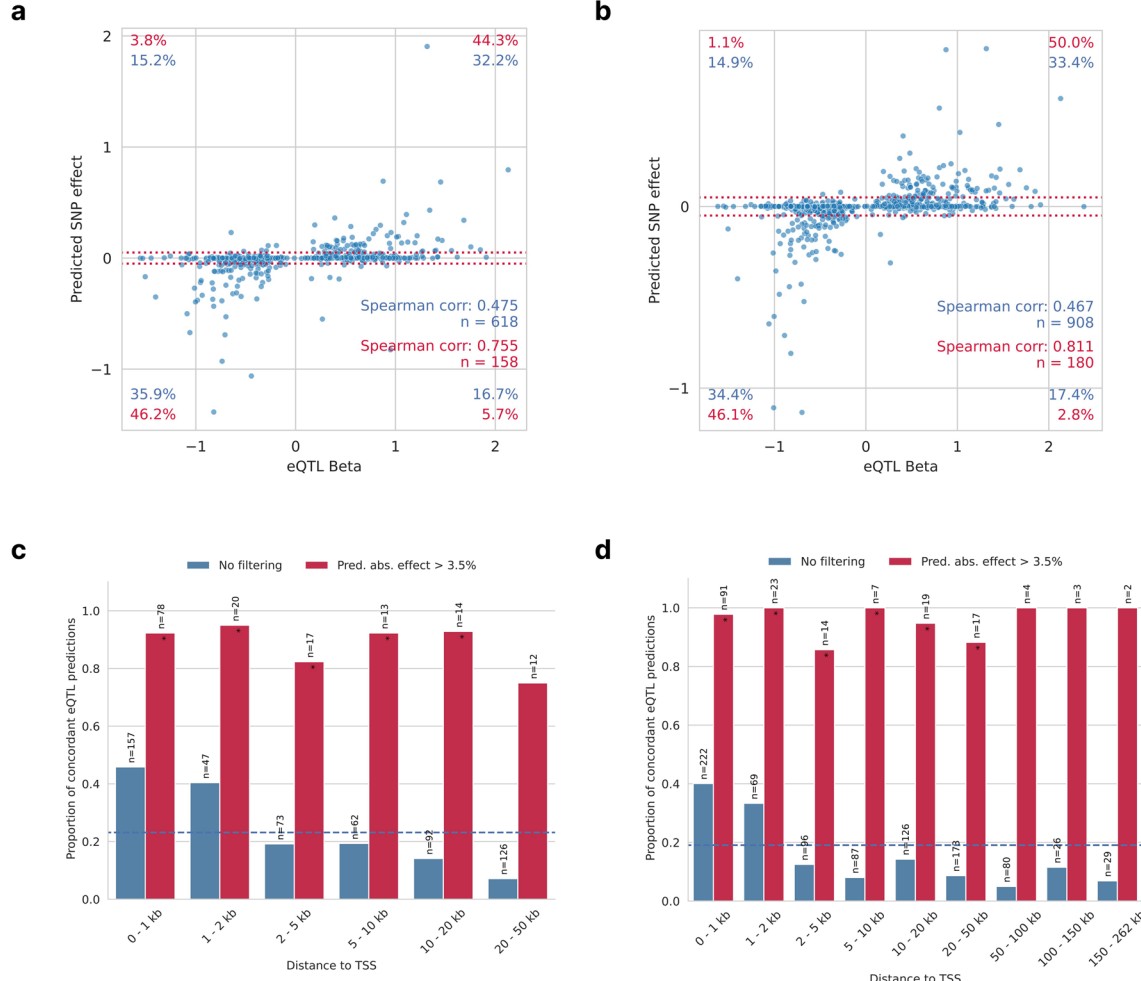

**Extended Data Fig. 9 | Variant effect prediction performance on GTEx whole blood eQTLs for seq2cells and Borzoi. a**, Seq2cells predicted aggregated effects (log-fold change) vs. observed whole-blood eQTL effect sizes. Red dotted lines mark thresholds below which predicted fold-changes are deemed negligible (absolute fold change 3.5%; matching the threshold by Schwessinger et al. for comparability). Percentages quantify variants within each quadrant: blue - all variants; red - variants passing the 3.5% predicted effect threshold. **b**, Same as in **a**, but for Borzoi. **c**, Proportion of concordant seq2cells eQTL predictions (same direction as observed), as a function of distance to the transcription start site when filtering for non-negligible predicted effect (red) or without filtering (blue). Dashed blue line indicates the mean proportion of concordant eQTL predictions across all distances (0.23). Stars indicate significance over random performance (Binomial test). **d**, Same as in **c**, but for Borzoi.

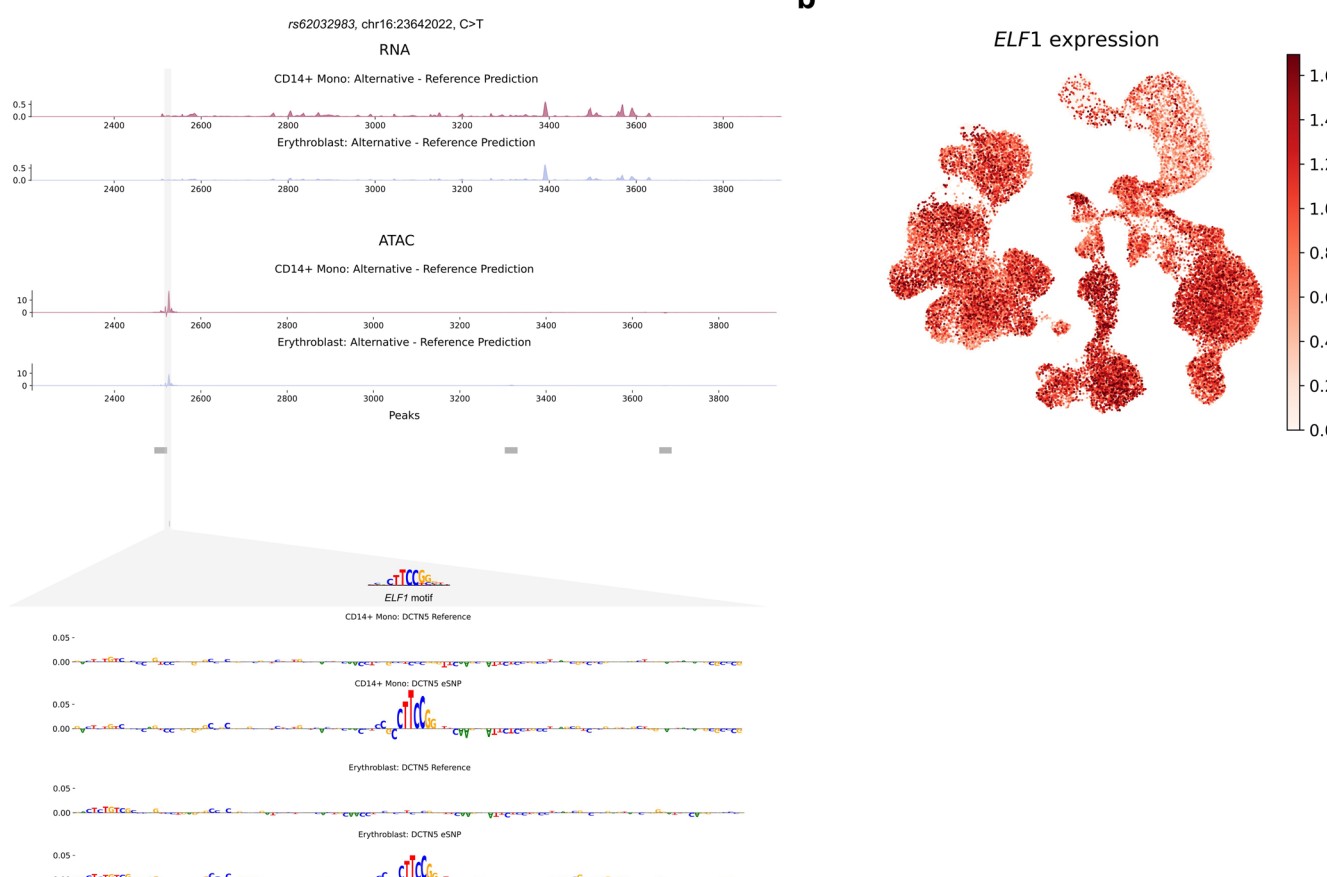

**a**

**b**

**Extended Data Fig. 10 | Example of a cell-type unspecific variant. a,** Predicted fold change in gene expression (top) and accessibility (bottom) between the alternative and reference alleles of variant rs62032983 in CD14+ Monocytes and Erythroblasts. Sequence attributions revealed the destruction of an *ELF1* motif to affect model outputs across cell types (Methods). **b,** UMAP of observed normalized *ELF1* expression levels.

# Reporting Summary

## Statistics

For all statistical analyses, confirm that the following items are present in the figure legend, table legend, main text, or Methods section.

| n/a | Confirmed | |
|---|---|---|
| ☐ | ☒ | The exact sample size (*n*) for each experimental group/condition, given as a discrete number and unit of measurement |
| ☒ | ☐ | A statement on whether measurements were taken from distinct samples or whether the same sample was measured repeatedly |
| ☐ | ☒ | The statistical test(s) used AND whether they are one- or two-sided *Only common tests should be described solely by name; describe more complex techniques in the Methods section.* |
| ☒ | ☐ | A description of all covariates tested |
| ☐ | ☒ | A description of any assumptions or corrections, such as tests of normality and adjustment for multiple comparisons |
| ☐ | ☒ | A full description of the statistical parameters including central tendency (e.g. means) or other basic estimates (e.g. regression coefficient) AND variation (e.g. standard deviation) or associated estimates of uncertainty (e.g. confidence intervals) |
| ☐ | ☒ | For null hypothesis testing, the test statistic (e.g. *F*, *t*, *r*) with confidence intervals, effect sizes, degrees of freedom and *P* value noted *Give P values as exact values whenever suitable.* |
| ☒ | ☐ | For Bayesian analysis, information on the choice of priors and Markov chain Monte Carlo settings |
| ☒ | ☐ | For hierarchical and complex designs, identification of the appropriate level for tests and full reporting of outcomes |
| ☐ | ☒ | Estimates of effect sizes (e.g. Cohen's *d*, Pearson's *r*), indicating how they were calculated |

*Our web collection on statistics for biologists contains articles on many of the points above.*

## Software and code

Policy information about availability of computer code

| Data collection | No software was used to collect data. |
|---|---|
| Data analysis | We used the following python packages with Python (v3.9.19): scanpy (v1.10), CellRanger ATAC (v2.1.0), Cell Ranger (v8.0.1), Cell Ranger (v6.1.1), pyRanges (v0.0.129), scvi (v1.1.2, https://github.com/lauradmartens/scvi-tools/tree/poissonmultivi), SnapATAC2 (v1.0.1, https://github.com/lauradmartens/SnapATAC2), rustup (v1.28.1), rustc (v1.85.0), scarches (v0.6.1), Borzoi (v0.0.2, https://github.com/johahi/borzoi-pytorch), peft (v0.10.1, https://github.com/lauradmartens/peft), trackplot (v0.4.0),  PyTorch (v2.1.0), tangermeme (v0.2.3), pychromvar (v0.0.4), seq2cells (https://github.com/GSK-AI/seq2cells), Unipressed  (v1.3.0), Meme suite, scipy(v1.13.1), gseapy (v1.1.3), seq2cells (https://github.com/GSK-AI/seq2cells), jupyterlab (v4.2.0), tomtom (v5.5.2). The scooby model including training scripts and data loaders are available at https://github.com/gagneurlab/scooby. Jupyter notebooks and scripts to reproduce our analysis and figures are available at https://github.com/gagneurlab/scooby_reproducibility.  The adapted version of SnapATAC2 is available at https://github.com/lauradmartens/SnapATAC2. The scooby model including training scripts and data loaders are available at https://github.com/gagneurlab/scooby. Jupyter notebooks and scripts to reproduce our analysis and figures are available at https://github.com/gagneurlab/scooby_reproducibility. The adapted version of SnapATAC2 is available at https://github.com/lauradmartens/SnapATAC2. The code along with data to reproduce the findings have additionally been archived and are available on Zenodo at https://doi.org/10.5281/zenodo.15517764 and https://doi.org/10.5281/zenodo.15517072. |

For manuscripts utilizing custom algorithms or software that are central to the research but not yet described in published literature, software must be made available to editors and reviewers. We strongly encourage code deposition in a community repository (e.g. GitHub). See the Nature Portfolio guidelines for submitting code & software for further information.

## Data

Policy information about availability of data

 All manuscripts must include a data availability statement. This statement should provide the following information, where applicable:
- Accession codes, unique identifiers, or web links for publicly available datasets
- A description of any restrictions on data availability
- For clinical datasets or third party data, please ensure that the statement adheres to our policy

The scRNA-seq, scATAC-seq, and pre-processed count matrices for the multiome hematopoiesis dataset are available from the NeurIPS 2021 challenge, SRA (accession SRP356158), AWS (s3://openproblems-bio/public/post_competition/multiome/), and GEO (accession GSE194122). The epicardioids dataset raw data (scATAC-seq, scRNA-seq) is available from SRA (accessions SRP359250, SRP359249). The OneK1K dataset raw data (scRNA-seq) is available from SRA (accession SRP359840). Pre-processed OneK1K data is available from CZ CELLxGENE: https://cellxgene.cziscience.com/collections/dde06e0f-ab3b-46be-96a2-a8082383c4a1. We used the Cell Ranger references refdata-cellranger-arc-GRCh38-2020-A-2.0.0 and refdata-gex-GRCh38-2020-A. We used the GENCODE release v32 GTF file and the GO Biological Process 2021 gene set. TF position weight matrices were obtained from HOCOMOCO v12 (https://hocomoco12.autosome.org/downloads_v12). GTEx eQTL and OneK1K summary statistics and fine-mapping results are available at https://www.ebi.ac.uk/eqtl/. The scATAC-seq PBMC dataset was downloaded from https://app.azimuth.hubmapconsortium.org/app/human-pbmc-atac. The GWAS Catalog (v1.0) was downloaded from https://www.ebi.ac.uk/gwas/docs/file-downloads. We matched SNP IDs using dbsnp (rs_id_dbSNP151_GRCh38p7, https://www.ncbi.nlm.nih.gov/snp/).

## Research involving human participants, their data, or biological material

Policy information about studies with human participants or human data. See also policy information about sex, gender (identity/presentation), and sexual orientation and race, ethnicity and racism.

| | |
|---|---|
| Reporting on sex and gender | n/a |
| Reporting on race, ethnicity, or other socially relevant groupings | n/a |
| Population characteristics | n/a |
| Recruitment | n/a |
| Ethics oversight | n/a |

Note that full information on the approval of the study protocol must also be provided in the manuscript.

# Field-specific reporting

Please select the one below that is the best fit for your research. If you are not sure, read the appropriate sections before making your selection.

☒ Life sciences ☐ Behavioural & social sciences ☐ Ecological, evolutionary & environmental sciences

For a reference copy of the document with all sections, see nature.com/documents/nr-reporting-summary-flat.pdf

# Life sciences study design

All studies must disclose on these points even when the disclosure is negative.

| | |
|---|---|
| Sample size | The study utilized three primary single-cell datasets (NeurIPS Hematopoiesis dataset, OneK1K Cohort, Heart Organoid dataset), supplemented by GTEx bulk eQTLs, to comprehensively evaluate scooby's diverse capabilities. Collectively, these datasets were sufficient as they enabled assessment across different biological contexts, scales, analytical tasks (profile modeling, TF activity, eQTL prediction), and allowed benchmarking against existing methods, supporting the paper's main claims. |
| Data exclusions | Peaks and genes from the datasets were excluded when they had counts in less than 1% of the cells. We identified and removed doublet cell populations in the Neurips dataset using Scrublet with default parameters. Doublet calls were based on a threshold that primarily captured cells clustering in discrete locations on the Uniform Manifold Approximation and Projection (UMAP) embedding. We removed the cell types 'Platelets' and 'Erythrocytes' to retain only immune cell types for the OneK1K dataset. For the epicardioids dataset we retained cells for which we had a scRNA-seq and scATAC-seq match. |
| Replication | We ensured robust evaluation by following the same sequence-level train and test splits as our underlying foundation model Borzoi. Moreover, genes and scATAC-seq peaks overlapping with validation or test regions were excluded from the input data used to generate the single-cell embeddings to avoid data leakage. No experimental findings were disclosed, hence no replication was performed. |
| Randomization | Random allocation is not relevant as this is a computational modeling study using pre-existing datasets. Unbiased evaluation is ensured through fixed train/validation/test splits of genomic data, preventing data leakage, even from the underlying Borzoi model. Scooby explicitly |

accounts for key covariates like cell type by conditioning predictions on cell-specific embeddings. Performance is then assessed with objective metrics, often stratified by these known biological factors, ensuring robust and fair evaluation.

Blinding | Blinding of investigators to group allocation during data collection was not relevant to this study. The research involves the development, training, and evaluation of a computational model (scooby) applied to pre-existing, publicly available single-cell multi-omics datasets (e.g., NeurIPS hematopoiesis dataset, OneK1K cohort, Heart organoid dataset)

# Reporting for specific materials, systems and methods

We require information from authors about some types of materials, experimental systems and methods used in many studies. Here, indicate whether each material, system or method listed is relevant to your study. If you are not sure if a list item applies to your research, read the appropriate section before selecting a response.

## Materials & experimental systems

| n/a | Involved in the study |
|-----|----------------------|
| ☒ ☐ | Antibodies |
| ☒ ☐ | Eukaryotic cell lines |
| ☒ ☐ | Palaeontology and archaeology |
| ☒ ☐ | Animals and other organisms |
| ☒ ☐ | Clinical data |
| ☒ ☐ | Dual use research of concern |
| ☒ ☐ | Plants |

## Methods

| n/a | Involved in the study |
|-----|----------------------|
| ☒ ☐ | ChIP-seq |
| ☒ ☐ | Flow cytometry |
| ☒ ☐ | MRI-based neuroimaging |

## Plants

| Seed stocks | n/a |
|-------------|-----|
| Novel plant genotypes | n/a |
| Authentication | n/a |

