## [Peer Review File · Nature Methods]

scooby: Modeling multi-modal genomic profiles from DNA sequence at single-cell resolution

Corresponding Author: Dr Julien Gagneur

Version 0:

Decision Letter:

26th Nov 2024

Dear Professor Gagneur,

Your Article, "scooby: Modeling multi-modal genomic profiles from DNA sequence at single-cell resolution", has now been seen by 2 reviewers. As you will see from their comments below, although the reviewers find your work of potential interest, they have raised a number of concerns. We are interested in the possibility of publishing your paper in Nature Methods, but would like to consider your response to these concerns before we reach a final decision on publication.

We therefore invite you to extensively revise your manuscript to address full these concerns.

Since scooby uses Borzoi which is presented in a preprint rather than a journal paper, please elaborate on this issue.

Link Redacted

We hope to receive your revised paper within 3 months. If you cannot send it within this time, please let us know. In this event, we will still be happy to reconsider your paper at a later date so long as nothing similar has been accepted for publication at Nature Methods or published elsewhere.

OPEN SCIENCE REQUIREMENTS

REPORTING SUMMARY AND EDITORIAL POLICY CHECKLISTS

Reporting summary: <https://www.nature.com/documents/nr-reporting-summary.zip>
Editorial policy checklist: <https://www.nature.com/documents/nr-editorial-policy-checklist.zip>

DATA AVAILABILITY

All novel DNA and RNA sequencing data, protein sequences, genetic polymorphisms, linked genotype and phenotype data, gene expression data, macromolecular structures, and proteomics data must be deposited in a publicly accessible database, and accession codes and associated hyperlinks must be provided in the "Data Availability" section.

CODE AVAILABILITY

Please include a "Code Availability" subsection in the Online Methods which details how your custom code is made available. Only in rare cases (where code is not central to the main conclusions of the paper) is the statement "available upon request" allowed (and reasons should be specified).

For more information on our code sharing policy and requirements, please see: <https://www.nature.com/nature-research/editorial-policies/reporting-standards#availability-of-computer-code>

MATERIALS AVAILABILITY

ORCID

Nature Methods is committed to improving transparency in authorship. As part of our efforts in this direction, we are now

requesting that all authors identified as 'corresponding author' on published papers create and link their Open Researcher and Contributor Identifier (ORCID) with their account on the Manuscript Tracking System (MTS), prior to acceptance. This applies to primary research papers only. ORCID helps the scientific community achieve unambiguous attribution of all scholarly contributions. You can create and link your ORCID from the home page of the MTS by clicking on 'Modify my Springer Nature account'. For more information please visit www.springernature.com/orcid.

Sincerely,

Lin Tang, PhD
Senior Editor
Nature Methods

Reviewers' Comments:

Reviewer #1 (Remarks to the Author):

scooby is a method proposed by Hingerl et al. for predicting RNA-seq and ATAC-seq profiles with cell type resolution given DNA sequence and an embedding of cell type. To encode DNA sequence, they use Borzoi as a foundation model off of which they perform parameter efficient finetuning. To encode cell type, the authors use a subset of the RNA-seq and ATAC-seq data (different from the data used in evaluation) to learn a joint embedding using MultiVI.

Overall, I found some of the downstream use cases around QTL interpretation interesting and compelling, but I had some major questions about (1) the presentation of the method as a predictive model that is meant to generalize out of its training distribution and (2) the presence of data leakage in the evaluations that are set up to assess this generalization.

Given these major questions, I do not think they are a barrier to publication, but I also think that I would like to hear the authors' answers, combined with major revisions, before I would be able to recommend publication of the manuscript.

Major questions

- What would be the theoretical performance of perfect prediction? For example, if one were to predict experimental single cell RNA-seq and ATAC-seq profiles using the experimental bulk profiles, what would the Spearman and Pearson correlations be? Having this value would help ground the overall significance of the results, and could also potentially help disentangle how much of the performance decrease with respect to Borzoi is due to the noise of single-cell data versus limitations with the model itself.
- I am not completely convinced that the authors have constructed an evaluation that completely prevents data leakage across different cell types. While the Borzoi train/test splits at the sequence/gene level are good, there is not very much methodological detail describing how the authors split their cell types. This is especially important given that the cell type information is encoded as MultiVI embeddings that appear to be constructed on the total dataset (both train and test). If so, then this is improper experimental procedure – if the goal of the model is to truly generalize to unseen cell types, then there is a high chance that a joint embedding across train/test can leak information across splits.
- Somewhat related to the above, I am skeptical about describing the model as a method that generalizes beyond its training dataset to new cell types. Leakage issues aside, the model is only trained on cell types from the bone marrow/hematopoietic lineage, so I would not expect these results to generalize to most cells in the brain or the lung, for example. The authors do add some qualifications in the discussion, but I am wondering if the overall presentation of the model can be shifted to de-emphasize it as a model used for prediction in the sense of new cell annotation, and rather as a model meant to simplify interpretation of cell types within its training distribution.
- I am also skeptical of describing the overall model architecture as a method for predicting RNA/ATAC peaks, especially because the cell type conditioning requires RNA/ATAC data at the cell type level. For example, a user who would want to go from DNA sequence to single-cell RNA/ATAC profile would also need to collect cell type specific RNA/ATAC data as additional cell type conditioning to the model in order to construct the cell type embedding – but at this point, the user has already run an RNA/ATAC assay with cell type resolution that would seemingly defeat the purpose of needing the scooby method (or at the very least, would substantially decrease the computational method's utility). Please let me know if this characterization is accurate.
- Lastly, given the above questions, it seems like the model is best suited for (re)interpretation of an existing scRNA/scATAC dataset (e.g., revisiting causal eQTL prediction or providing cell type specific interpretation of bulk eQTLs) rather than as a predictive model that needs to generalize out-of-distribution to new cell types or to different RNA/ATAC data subsets. I do think that these tasks are interesting and important. However, this also means that the authors should do a more thorough job benchmarking these aspects of the model using simple controls (e.g., what if we used RNA expression/ATAC peak height),

rather than scooby, to do causal eQTL prediction. The authors should provide these simple statistical baselines, as well as comparisons to state-of-the-art models, in these tasks that are relevant to a downstream user. As is, I feel that the authors comparison only to seq2cells is insufficient.

Reviewer #1 (Remarks on code availability):

I was able to install the PyPI package and run a very basic test.

Reviewer #2 (Remarks to the Author):

In this manuscript, the authors present a deep learning model (Scooby) that predicts single-cell multi-omic coverages (ATAC+RNA) from DNA sequences and single-cell embeddings. To extract sequence embeddings, the model uses LoRA (low-rank adaptation) to fine-tune a previously published model, Borzoi, which takes ~500kb sequences and predicts bulk RNA-Seq coverages. Scooby then uses a lightweight cell-specific decoder to combine single-cell embeddings extracted from Multi-VI with the sequence embeddings to predict cell-specific coverages (ATAC and RNA). While the manuscript is well-written and clear, with well-documented code and technically sound analyses, I still have some concerns regarding methodological choices and the novelty of the method. My detailed comments are below:

Major Comments:

1. Scooby's key advantage over other single-cell seq2function models is its potential to use any cell embedding extracted from single-cell datasets to predict single-cell coverage, theoretically generalizing to unseen cell types/states. However, the authors demonstrate this with only one withheld cell type, which is insufficient to convincingly show generalization across diverse cell types, especially those from different lineages or datasets.
2. While finetuning Borzoi is a reasonable approach, I wonder whether this is needed. A simpler alternative the authors should compare to is using all or the RNA-seq subset of Borzoi's predictions as an encoding.
3. The benefits of using cell embeddings as additional input are not clearly demonstrated. There is no direct comparison to a baseline model. The authors should consider directly fine-tuning Borzoi to predict single-cell coverage from sequences without cell embeddings and compare it with Scooby.
4. While the authors claim Scooby's predictions are at the single-cell level, it was not systematically examined whether Scooby's predictions can reflect biological single-cell heterogeneity within a cell type.
5. It appears that the eQTL effect prediction performance is much improved over Borzoi. However, a direct comparison is lacking. If there is substantial improvement, it will be very helpful to figure out what leads to the improvement. Since it would be a surprising result given that Scooby is based on Borzoi, this effect deserves more in-depth study.

Minor Comment:

It will add to the contribution of this manuscript if the authors can discuss novel biology discovered or hypotheses generated by the model.

Version 1:

Decision Letter:

Our ref: NMETH-A57946A

22nd Apr 2025

Dear Dr. Gagneur,

Thank you for submitting your revised manuscript "scooby: Modeling multi-modal genomic profiles from DNA sequence at single-cell resolution" (NMETH-A57946A). It has now been seen by the original referees and their comments are below. The reviewers find that the paper has improved in revision, and therefore we'll be happy in principle to publish it in Nature Methods, pending minor revisions to comply with our editorial and formatting guidelines.

TRANSPARENT PEER REVIEW

Nature Methods offers a transparent peer review option for new original research manuscripts submitted from 17th February 2021. We encourage increased transparency in peer review by publishing the reviewer comments, author rebuttal letters and

editorial decision letters if the authors agree. Such peer review material is made available as a supplementary peer review file. **Please state in the cover letter 'I wish to participate in transparent peer review' if you want to opt in, or 'I do not wish to participate in transparent peer review' if you don't.** Failure to state your preference will result in delays in accepting your manuscript for publication.

ORCID

Sincerely,

Lin Tang, PhD
Senior Editor
Nature Methods

Reviewer #1 (Remarks to the Author):

The authors have adequately responded to my review and I am happy to recommend acceptance of the paper.

Reviewer #2 (Remarks to the Author):

I appreciate the authors' revisions which have improved the manuscript. I congratulate the authors on their contribution.

Reviewer #2 (Remarks on code availability):

The repos are well organized.

Version 2:

Decision Letter:

28th Aug 2025

Dear Dr Gagneur,

I am pleased to inform you that your Article, "scooby: Modeling multi-modal genomic profiles from DNA sequence at single-cell resolution", has now been accepted for publication in Nature Methods. The received and accepted dates will be 18th Sep 2024 and 28th Aug 2025. This note is intended to let you know what to expect from us over the next month or so, and to let you know where to address any further questions.

Over the next few weeks, your paper will be copyedited to ensure that it conforms to Nature Methods style. Once your paper is typeset, you will receive an email with a link to choose the appropriate publishing options for your paper and our Author Services team will be in touch regarding any additional information that may be required. It is extremely important that you let us know now whether you will be difficult to contact over the next month. If this is the case, we ask that you send us the contact information (email, phone and fax) of someone who will be able to check the proofs and deal with any last-minute problems.

Authors may need to take specific actions to achieve compliance with funder and institutional open access mandates.

If your research is supported by a funder that requires immediate open access (e.g. according to [Plan S principles](https://www.springernature.com/gp/open-science/plan-s-compliance) or the [NIH public access policy](https://www.springernature.com/gp/open-science/us-federal-agency-compliance)) then you should select the gold OA route, and we will direct you to the compliant route where possible. Because authors warrant under our subscription licensing terms that they haven't committed to licensing any version of their article under a licence inconsistent with the terms of our agreement – including the applicable embargo period – publication under the subscription model isn't suitable for authors whose funders require no embargo.

Please feel free to contact me if you have questions about any of these points. Thank you very much for publishing your paper at Nature Methods!

Best regards,

Lin Tang, PhD
Senior Editor
Nature Methods

** Visit the Springer Nature Editorial and Publishing website at http://editorial-jobs.springernature.com?utm_source=ejp_NMeth_email&utm_medium=ejp_NMeth_email&utm_campaign=ejp_Nmeth for more information about our career opportunities. If you have any questions please click [here](mailto:editorial.publishing.jobs@springernature.com).**

Reviewers' Comments:

Reviewer #1 (Remarks to the Author):

scooby is a method proposed by Hingerl et al. for predicting RNA-seq and ATAC-seq profiles with cell type resolution given DNA sequence and an embedding of cell type. To encode DNA sequence, they use Borzoi as a foundation model off of which they perform parameter efficient finetuning. To encode cell type, the authors use a subset of the RNA-seq and ATAC-seq data (different from the data used in evaluation) to learn a joint embedding using MultiVI.

Overall, I found some of the downstream use cases around QTL interpretation interesting and compelling, but I had some major questions about (1) the presentation of the method as a predictive model that is meant to generalize out of its training distribution and (2) the presence of data leakage in the evaluations that are set up to assess this generalization.

Given these major questions, I do not think they are a barrier to publication, but I also think that I would like to hear the authors' answers, combined with major revisions, before I would be able to recommend publication of the manuscript.

Major questions

1. What would be the theoretical performance of perfect prediction? For example, if one were to predict experimental single cell RNA-seq and ATAC-seq profiles using the experimental bulk profiles, what would the Spearman and Pearson correlations be? Having

this value would help ground the overall significance of the results, and could also potentially help disentangle how much of the performance decrease with respect to Borzoi is due to the noise of single-cell data versus limitations with the model itself.

We thank the reviewer for highlighting the importance of establishing theoretical performance limits in single-cell data. We have reworked Fig 1d and the corresponding text to address this issue.

First, we are no longer providing the correlations of Borzoi with bulk RNA-seq and DNase-seq as these assays differ in many ways from scRNA-seq and scATAC-seq and are therefore not directly comparable. Following the reviewer's suggestion, we now provide the correlations between pseudobulk and single-cell profiles. Remarkably, scooby predictions fit the single-cell observations closer than the pseudobulks. We proposed as an alternative ground truth proxy the profile averaged over the 100 nearest neighbors. Scooby's agreement with single-cell profiles lies between these two. We hope that this better contextualizes scooby's performance within the inherent challenges of modeling single-cell data.

2. I am not completely convinced that the authors have constructed an evaluation that completely prevents data leakage across different cell types. While the Borzoi train/test splits at the sequence/gene level are good, there is not very much methodological detail describing how the authors split their cell types. This is especially important given that the cell type information is encoded as MultiVI embeddings that appear to be constructed on the total dataset (both train and test). If so, then this is improper experimental procedure – if the goal of the model is to truly generalize to unseen cell types, then there is a high chance that a joint embedding across train/test can leak information across splits.

While we ensured that the MultiVI embedding was generated without using any genomic sequences from the test set, it used all cells. Therefore, the generalization to held-out cell types (but not to held-out sequences) could have been overestimated. To address this, we have now redone the normoblast experiment with completely excluded normoblast cells. Specifically, we trained a new version of scooby where normoblast cells were not used during the training of the embeddings nor of scooby (only used after training by projecting them into the learned MultiVI space). This ensures that the model is truly tested on cells and cell states not seen during training. The updated analysis demonstrates that this modification does not affect scooby's performance on unseen normoblast gene expression prediction (updated Figure 2F, new correlation = 0.79, formerly = 0.78) or in modeling continuous expression patterns (updated Figure 2G, new Pearson R 0.966, formerly = 0.937).

3. Somewhat related to the above, I am skeptical about describing the model as a method that generalizes beyond its training dataset to new cell types. Leakage issues aside, the model is only trained on cell types from the bone marrow/hematopoietic lineage, so I would not expect these results to generalize to most cells in the brain or the lung, for example. The authors do add some qualifications in the discussion, but I am wondering if the overall presentation of the model can be shifted to de-emphasize it as a model used for prediction in the sense of new cell annotation, and rather as a model meant to simplify interpretation of cell types within its training distribution.

We completely agree with the reviewer's characterization: We do not expect scooby to generalize to drastically different cell types from the ones it was trained on, scooby is meant to be used for the interpretation of sequence determinants. The intended purpose of the normoblast experiment was to show scooby's robustness to slight changes in cell state in order to highlight its potential applicability to settings where we embed similar cell states post training, for instance by mapping query datasets to a reference atlas. We acknowledge that the presentation of these results was not explicit enough and could lead to misinterpretation. Therefore, we have revised our manuscript to de-emphasize scooby as a predictor generalizing to unseen cell types. Instead, we focus on its ability to model and interpret regulatory mechanisms within its training distribution. Along these lines, we also now favor the word "modeling" over "predicting" where applicable to emphasize that the application of scooby requires both DNA sequence and single-cell omics data (see points below).

4. I am also skeptical of describing the overall model architecture as a method for predicting RNA/ATAC peaks, especially because the cell type conditioning requires RNA/ATAC data at the cell type level. For example, a user who would want to go from DNA sequence to single-cell RNA/ATAC profile would also need to collect cell type specific RNA/ATAC data as additional cell type conditioning to the model in order to construct the cell type embedding – but at this point, the user has already run an RNA/ATAC assay with cell type resolution that would seemingly defeat the purpose of needing the scooby method (or at the very least, would substantially decrease the computational method's utility). Please let me know if this characterization is accurate.

We agree with the reviewer's characterization: scooby models single-cell omics data from DNA sequence and requires both as input to be applicable. Therefore, application to various cell types requires scATAC-seq and scRNA-seq from those cell types. To reflect this, we have reworded several sections throughout the manuscript to emphasize

modeling and interpretation rather than prediction. We are happy to improve this distinction further if the reviewer could point us to precise sections that are still misleading.

To be precise, we would like to highlight that scooby conditions at the single cell level, not at pseudobulked cell-type level, and we use the cell embedding vectors as an input to the model to give information about similarities between cells. In our manuscript, we therefore tried to consistently use the term “cell state” to describe similarly regulated cells and only use “cell type” when we pseudobulk the predictions using the coarse cell-type labels. This way of modeling distinguishes us from pseudobulk models and allows us to make interpretations on the cell state level as shown for example in the revised Figure 3f-i, where we show that scooby can find variations **within** a cell type.

5. Lastly, given the above questions, it seems like the model is best suited for (re)interpretation of an existing scRNA/scATAC dataset (e.g., revisiting causal eQTL prediction or providing cell type specific interpretation of bulk eQTLs) rather than as a predictive model that needs to generalize out-of-distribution to new cell types or to different RNA/ATAC data subsets. I do think that these tasks are interesting and important. However, this also means that the authors should do a more thorough job benchmarking these aspects of the model using simple controls (e.g., what if we used RNA expression/ATAC peak height), rather than scooby, to do causal eQTL prediction. The authors should provide these simple statistical baselines, as well as comparisons to state-of-the-art models, in these tasks that are relevant to a downstream user. As is, I feel that the authors comparison only to seq2cells is insufficient.

We have now implemented several additional analyses (revised Fig. 4, Extended Data Fig. 9):

- Our former eQTL benchmark was based on GTEx whole blood while scooby was trained on a bone marrow dataset. We now also trained scooby on a large single-cell PBMC dataset (OneK1K, 1.2 M cells from 982 donors), which better resembles whole blood than bone marrow. Moreover, the OneK1K dataset advantageously provides cell-type eQTLs ground truth, allowing for assessing cell-type specific eQTL performance.
- We now include two simple statistical baselines for the cell-type specificity prediction of eQTLs. Specifically, for a given eQTL we rank the cell-types according to i) the expression level of the eGene in the cell types, or ii) the chromatin accessibility of the closest ATAC peak to the eQTL in the cell types.

The results are shown in the completely revised Figure 4f. We found that for predicting effects in the cell types, these simple ranking strategies perform well but worse than scooby's fold-change predictions.

- Moreover, we now additionally benchmark against the state-of-the-art bulk expression model Borzoi. We find that scooby outperforms seq2cells and a track-matched Borzoi model on cell-type-specific eQTL prediction, while its performance on GTEx whole blood eQTLs nearly matches that of Borzoi. We integrate these results in Figure 4a,b.

Reviewer #1 (Remarks on code availability):

I was able to install the PyPI package and run a very basic test.

Reviewer #2 (Remarks to the Author):

In this manuscript, the authors present a deep learning model (Scooby) that predicts single-cell multi-omic coverages (ATAC+RNA) from DNA sequences and single-cell embeddings. To extract sequence embeddings, the model uses LoRA (low-rank adaptation) to fine-tune a previously published model, Borzoi, which takes ~500kb sequences and predicts bulk RNA-Seq coverages. Scooby then uses a lightweight cell-specific decoder to combine single-cell embeddings extracted from Multi-VI with the sequence embeddings to predict cell-specific coverages (ATAC and RNA). While the manuscript is well-written and clear, with well-documented code and technically sound analyses, I still have some concerns regarding methodological choices and the novelty of the method. My detailed comments are below:

Major Comments:

1. Scooby's key advantage over other single-cell seq2function models is its potential to use any cell embedding extracted from single-cell datasets to predict single-cell coverage, theoretically generalizing to unseen cell types/states. However, the authors demonstrate this with only one withheld cell type, which is insufficient to convincingly show generalization across diverse cell types, especially those from different lineages or datasets.

We acknowledge the reviewer's point regarding the limited evidence for generalization across diverse cell types. As also noted in our response to Reviewer 1, the main purpose of scooby is modeling, interpreting, and interpolating regulatory mechanisms and variant effects within the cell types and states present in a single dataset. We have revised our manuscript to de-emphasize the prediction of completely unrelated cell types, and instead highlight the mentioned capabilities even more. Our primary goal with the normoblast

ablation study was to demonstrate the robustness of the cell-specific decoder and to probe whether the learned embedding space was being effectively leveraged, rather than to showcase generalization to truly novel cell types or datasets. We foresee scooby to be trained either on individual single-cell datasets, or on large atlas-scale reference datasets. In the latter case, a robust cell-state-specific decoder could allow scooby to be applied to smaller datasets mapped to the reference. To show the generalizability of scooby to diverse and large-scale datasets, we have now trained and successfully applied scooby to two new datasets (heart organoids, OneK1K PBMC with 1.2M cells).

2. While finetuning Borzoi is a reasonable approach, I wonder whether this is needed. A simpler alternative the authors should compare to is using all or the RNA-seq subset of Borzoi's predictions as an encoding.

We had already trained an ablated model using only the Borzoi embeddings without fine-tuning, which performed significantly worse (Extended Data Fig. 5), but forgot the direct reference in the main text. This is now fixed. As suggested, we additionally trained a scooby model on all or the RNA-seq subset of Borzoi's predictions. These two models performed significantly worse than the aforementioned ablation on the non fine-tuned Borzoi embeddings (revised Extended Data Fig 5). Of note, we encountered unstable training due to the large numerical range of the original Borzoi output tracks, so that we had to resort to lower learning rates for these two models.

3. The benefits of using cell embeddings as additional input are not clearly demonstrated. There is no direct comparison to a baseline model. The authors should consider directly fine-tuning Borzoi to predict single-cell coverage from sequences without cell embeddings and compare it with Scooby.

We acknowledge the reviewer's point about the need to benchmark scooby against a model without cell embeddings. We considered fine-tuning the Borzoi model directly to predict single-cell coverage from sequence without cell embeddings. While computationally feasible for a small dataset, this approach becomes fundamentally unscalable for larger datasets. Training such a model on our dataset, where each training sequence would require predicting profiles for ~60,000 cells, results in prohibitive GPU memory requirements for a single forward and backward pass. To illustrate, target data alone for such a model would necessitate a memory footprint of 4.5 GB per training sequence (60,000 cells x 6144 bins x 3 assays x 4 bytes (float32)). By contrast, scooby, with its shared cell-state-specific decoder, avoids this parameter explosion: For each training sequence, scooby learns how to decode the sequence embedding into a profile,

and it conditions this decoding process on a low-dimensional cell embedding that represents the cell's state. Therefore, during training, scooby only needs to process profiles for a representative subset of cells (e.g., 128 cells in our setup) for each sequence, because the decoder parameters are shared and learn to generalize across the entire cell-state space. This parameter-efficient design is crucial for scooby's scalability and enables scooby to scale to the OneK1K dataset (with 1.2 million PBMC cells), where even count-based approaches like seq2cells become intractable and require data subsampling (Methods).

4. While the authors claim Scooby's predictions are at the single-cell level, it was not systematically examined whether Scooby's predictions can reflect biological single-cell heterogeneity within a cell type.

We thank the reviewer for raising the important question of whether scooby's predictions can capture biological heterogeneity within a cell type, rather than just between distinct cell types. To investigate this, we analyzed a published single-cell dataset of human heart organoids, a model system experimentally validated to exhibit heterogeneity within its progenitor cell population, the juxta-cardiac field progenitors (JCFs). Specifically, the heart organoid dataset captures two substates of JCF progenitor cells that differentiate into either cardiomyocytes (CMs) or epicardial cells with previously characterized lineage-specific TF signatures, including GATA4 for CM differentiation and FOS for epicardial differentiation. Using scooby's in silico motif deletion analysis, we found that FOS motifs exhibit the strongest effect in JCFs committed to the epicardioid lineage, while GATA4 motifs have strong effects in JCFs that are committed towards the cardiomyocyte lineage (revised Figure 3f-i). This demonstrates scooby's ability to uncover cell-state specific regulation and single-cell heterogeneity even within a cell type. We extended the manuscript with these findings.

5. It appears that the eQTL effect prediction performance is much improved over Borzoi. However, a direct comparison is lacking. If there is substantial improvement, it will be very helpful to figure out what leads to the improvement. Since it would be a surprising result given that Scooby is based on Borzoi, this effect deserves more in-depth study.

In our original manuscript, we had made some qualitative comparisons in the discussion about eQTL sign concordance or lack thereof reported by other studies. However, this part of the text was probably confusing as we had not performed any direct comparison against Borzoi. We have now integrated a direct comparison with Borzoi. We did this in two ways. On the one hand we used aggregated predictions across all OneK1K cell types

for scooby to compare to whole-blood predictions of Borzoi. On the other hand, we identified prediction tracks of Borzoi that could be matched to cell types present in the single-cell PBMC dataset. As mentioned in our response to reviewer 1, point 5, we find that Borzoi is mildly better than scooby on GTEx whole blood eQTL effect prediction (Spearman = 0.466 vs 0.449). The advantage of scooby is to bring the single-cell resolution. On the cell-type specific eQTLs from the OneK1K cohort, scooby outperforms the track-matched Borzoi (new Fig. 4) on all cell-types.

Moreover, we have now investigated Borzoi regarding sign concordance and found it to behave similarly to scooby (new Extended Data Fig. 9). We changed the discussion paragraph to reflect this observation.

Minor Comment:

It will add to the contribution of this manuscript if the authors can discuss novel biology discovered or hypotheses generated by the model.

As mentioned in the response to point 4, we have added an analysis of human heart organoid data using scooby. We have included a vignette outlining how the motif effect scores can be used to identify putative transcription factors that may regulate JCF lineage commitment towards cardiomyocyte or epicardial fates.